

**Fracture attribute scaling and connectivity in the Devonian Orcadian**
**Basin with implications for geologically equivalent sub-surface**
**fractured reservoirs**
Anna M. Dichiarante[1,2], Ken J.W. McCaffrey[1,3], Robert E. Holdsworth[1,3], Tore I. Bjornarå[4] and
Edward D. Dempsey[5]
*[1] Department of Earth Sciences, Durham University, Durham DH1 3LE, UK*
*[2] NORSAR, Kjeller, Norway*
*[3] Geospatial Research Ltd, 1 Hawthorn Terrace, Durham, DH1 4EL, UK*
*[4] NGI - Norges Geotekniske Institutt*
*[5] Department of Geography, Geology and Environment, University of Hull, Hull HU6 7RX, UK*
*Correspondance to: k.j.w.mccaffrey@durham.ac.uk*





**Abstract**: Fracture attribute scaling and connectivity datasets from analogue systems are widely used
to inform sub-surface fractured reservoir models in a range of geological settings. However,
significant uncertainties are associated with the determination of reliable scaling parameters in
surface exposures, particularly for fault widths and fracture aperture. This has limited our ability to
upscale key parameters that control fluid-flow at reservoir to basin scales. In this study, we present
nine 1D-transect (scanline) fault and fracture attribute datasets from Middle Devonian sandstones in
Caithness (Scotland) that are widely used as an onshore analogue for nearby sub-surface reservoirs
such as the Clair Field, West of Shetland. Our multiscale analysis confirms power-law behaviours for
both length over 8 orders of magnitude ($10^{-4}$ to $10^{4}$) and fracture aperture and fault width (including
fills) over 4 orders of magnitude ($10^{-6}$ to $10^{-2}$). We also present a 2D fault and fracture topology
analysis which allows assessment of the heterogeneity of connectivity and self-similarity. This
multiscale approach provides a new basis for upscaling micro- to meso-scale fracture attributes
collected in outcrop analogues for use in static and dynamic reservoir models at reservoir to basin
scales.
*Keywords*: fault, fracture attribute, multiscale, Clair Field, Devonian





## 1 Introduction

Fractures, used here as a general term to include faults, joints and veins, exist over a wide range of scales from microns to hundreds of kilometres and fundamentally control the fluid-flow and mechanical properties of crustal rocks, including many sub-surface reservoirs holding oil, gas or water (e.g. Sibson, 1996; Odling et al., 1999; Bonnet et al., 2001). The heterogeneous distribution of natural fracture systems and uncertainties associated with the determination of reliable scaling parameters in 3D over large-scale ranges remains a persistent problem in reservoir studies. Small faults (fractures with <1m displacement) and joints (fractures with no shear displacement) may occur in isolation, or as part of a damage zone of larger displacement (>10s m) faults (e.g. Shultz and Fossen, 2008). Fractures (*sensu lato*) can be described by their *geometrical* attributes such as orientation and size (displacement, length, aperture) and also by *spatial* attributes such as intensity (density), clustering, connectivity and continuity (Sanderson and Nixon, 2015).

In practice, fracture attribute distributions are typically constrained from drillholes and cores which provide high resolution ($10^{-4}$ to $10^{0}$ m), but highly censored (size limited by borehole diameter), spatially limited 1D samples of the reservoir. In contrast, 2D and 3D seismic reflection datasets provide continuous, but relatively low-resolution fracture network maps at 10 to $10^{5}$ m scales. Consequently, since fluid-flow in a fractured reservoir is a volumetric (3D) summation of all small- (< 1m displacement) to large-scale fracture contributions, accurately characterizing 3D fracture network using just borehole-, cores- and seismic-derived datasets is particularly challenging.

Analogue datasets from outcrops are used to fill the gap between high-resolution but sparse borehole data and low-resolution seismic data and give access to fracture datasets across many scales ($10^{-2}$ to $10^{6}$ m scales) and in 1, 2 and 3 dimensions (e.g. Mäkel, 2007). Using one dimensional sampling methods (1D), fracture attributes in outcrop analogues have been investigated in different tectonic contexts and lithologies at given scales (e.g. Baecher, 1983; Gillespie et al., 1993; McCaffrey and Johnston, 1996; Knott et al., 1996; Odling et al., 1999; Bour et al., 2002; Manzocchi, 2002; Olson, 2003; Kim and Sanderson, 2005; Gomez and Laubach, 2006; Schultz et al., 2008; Hooker et al., 2009; Torabi and Berg, 2011). However, results from *multi-scale* sampling of fracture attributes are less common (e.g. Walsh and Watterson, 1988; Guerriero et al., 2010; Torabi and Berg, 2011; Bertrand et al., 2015).





In this study, we use a multi-scale 1D sampling approach to describe fractures (including faults)
formed in Devonian sandstones of the Orcadian Basin, North Scotland. These rocks are widely
viewed as being a useful analogue for the fractured Devonian siliclastic reservoirs that form the giant
Clair Field, West of Shetland (Allen and Mange-Rajetzky, 1992; Coney et al., 1993; Barr et al., 2007),
one of the largest remaining oilfields in the UKCS (c. 7 billion barrels of Stock Tank oil initially in
place, Robertson et al. 2020). We investigate here the size and spatial distributions of fracture trace
length (multi-scale) and aperture (small-scale only), and their scaling properties. In addition, we use
a multi-scale 2D sampling approach that allows us to quantify the connectivity of fracture networks
following the fracture topology methodology used by Nixon (2013) and Sanderson and Nixon (2015).
We use datasets derived from a high-resolution bathymetric map (sub-regional scale), aerial
photographs, coastal exposure and a thin-section made from hand samples. We discuss how this
integrated approach can be used to upscale analogue datasets - in particular the aperture/fracture width
parameter - to reservoir or basin scales. Whilst, we do not aim here to revise the theoretical statistical
background of fitting methods (extensively treated in Corral and González, 2019) we do show their
potential when applied to a multiscale analysis. This work shows that the determination of multiscale
fracture attributes scaling in 1D and 2D can form a useful input for building realistic static geological
models at reservoir scale that serve as starting points for simulations of fluid storage, migration
processes and production (e.g. Odling et al., 1999).

**2      Methodology**
2.1      Sampling of fractures and fracture network attributes
The most common data acquisition methodologies use: (i) scanlines (or transects); (ii) window
sampling; (iii) circular scanline windows; and (iv) box counting (**Fig. 1**a), which collectively provide
access to different attributes as shown in **Table 1**. Scanlines (1D method) allow a relatively simple
characterization of individual fracture sizes and spacing (**Fig. 1**b), and act as a good proxy for the
borehole data that typically serve as starting points for building reservoir models (Priest and Hudson,
1981; Baecher, 1983; Gillespie et al., 1993; McCaffrey and Johnston, 1996; Knott et al., 1996; Ortega
and Marrett, 2000; Ortega et al., 2006; Bonnet et al., 2001; Odling et al., 1999). Window sampling
and circular scanline windows (both 2D methods) provide further information on the spatial
relationships within the fractured system (Mauldon, 1994; Mauldon et al., 2001; Rohrbaugh et al.,
2002; Manzocchi, 2002; Zeeb et al., 2013; Watkins et al., 2015; Sanderson and Nixon, 2015; Rizzo
et al., 2017) and importantly provide access to connectivity estimates for the fracture array, which is
a key input when modelling fluid-flow.



In this study, fracture orientations, trace lengths and apertures, together with composition and
texture of fracture infills and fracture terminations on joints and faults were recorded. The start and
end point of each transect was recorded using a hand-held GPS unit. Most fractures in the Orcadian
Basin are filled with minerals (calcite or pyrite) or, locally, oil, and, following Ortega et al. (2006),
the apertures measured in this study are the orthogonal distance between the fracture walls and include
the fill, i.e. the *'kinematic aperture'*. When it was not possible to measure the transect orthogonally
to the main fault because of outcrop exposure limitations (e.g. at the sub-regional scale), the measured
attributes were adjusted using the Terzaghi's Correction (**Fig. 1**c). To more precisely measure the
aperture attributes, an aperture size comparator ($10^{-5}$ to $10^{-4}$m) was used in the field to in order to
ensure a larger range of recorded apertures, thereby reducing censoring effects.
To extend the analysis to other scales, the above mentioned scanline method was adapted and
applied to both aerial photographs (regional scale) to quantify trace length, and to thin-section
(microscale) to quantify trace length and aperture.

### 2.1.1   *Fracture intensity/frequency plots (1D)*
The fracture intensity/frequency distribution for 1D datasets can be visualised by plotting sorted
attribute values (e.g. fracture length) versus cumulative frequency. This enables assessment of the
distribution, spatial and scaling properties of the fracture sample (i.e. the ratio of short to long
fractures for given sample line length). Fracture attribute distributions are thought to display three
main types of statistical distribution (**Fig. 2**; Bonnet et al., 2001; Gillespie et al., 1993; Zeeb et al.,
2013): (a) *Exponential, random or Poisson distributions* are characteristic of a system with one
randomised variable (Gillespie et al., 1993); (b) *Log-normal distributions* are generally produced by
systems with a characteristic length scale, due to mechanical stratigraphic boundaries controlling
layer-bound jointing for example (Narr, 1991 and Olson, 2007); (c) *Power-law distributions* lack a
characteristic length scale in the fracture growth process (Zeeb et al., 2013) and the data exhibit scale-
invariant fractal geometries (**Fig. 2**c bottom). This means that the relative number of small versus
large elements remains the same at all scales between the upper and lower fractal limits (Barton,
1995). Limits to the fractal behaviour, although unknown (Corral and González, 2019), can be related
to both spatial and temporal influence, e.g., lithological boundaries across which fracture
characteristics change, and changes in stress orientation, respectively. Although some fracture
populations are better described by scale-limited laws, such as log-normal or exponential
distributions, it is generally accepted that for many systems power-law distributions and fractal



geometry provide a widely applicable descriptive tool for fracture system characterization (e.g.
Bonnet et al., 2001). Ideally, the best-fit in a power-law distribution should be consistent over several
orders of magnitude at a given scale (Walsh and Watterson, 1993; McCaffrey and Johnston, 1996).

Fracture sampling issues (e.g. censoring and truncation in **Fig. 2**c) are commonly encountered
and can result in difficulty in ascribing the best-fit distribution. For instance, when long fractures are
incompletely sampled (e.g. censoring in **Fig. 2**c), it is difficult to determine between log-normal and
power-law fits to distributions. These sampling issues (due to resolution effects) may mean that, while
a log-normal distribution is the best-fit to a dataset, a power-law distribution can also show a good fit
(Corral and González, 2019) and may be preferred because of its greater physical significance and
practical applicability (Bonnet et al., 2001). These assumptions need to be examined closely in any
analysis of scaling (see Clauset et al., 2009). The maximum likelihood estimator (MLE) is a statistical
technique that determines which distribution model is most likely to describe the data and it returns
governing parameters of the fitting equations (see Supplementary Data File). The Kolmogorov-
Smirnoff (KS) test is then used to evaluate the difference between the data and synthetic data
generated using the governing parameter derived from the MLE (Clauset et al., 2009). We use these
statistical methods and adapted the methodology proposed by Rizzo et al. (2017) and used by
FracPaQ (Healy et al., 2017) to calculate the MLE on progressively truncated populations for power-
law, exponential and log-normal distributions.

### 2.1.2   2D topology analysis

Whilst 1D analyses provide information about fractures as single entities and their distribution per
unit length of sample, 2D analyses measure fracture network properties and provide estimates of
fracture connectivity and self-similarity. The 2D analysis used here was carried out on fractures at
mesoscale using outcrop pavement photographs and at a larger scale using an offshore bathymetric
data. Circular scanline windows and box counting methods were performed using the Corel Draw
Graphic Suite™, ArcGis™ and MATLAB™ to produce small-scale fracture density maps  (**Fig. 2**d),
self-similarity plots (**Fig. 2**f) and ternary plots (**Fig. 2**e). To understand fracture topology, we follow
Sanderson and Nixon (2015) in considering that fracture arrays are typically composed of nodes and
branches. Nodes are points where a fracture terminates (I-type), abuts against another fracture
(Y-type) or intersects another fracture (X-type) and branches are the portions of a fracture confined
between two nodes. These are defined as I-I type (isolated branch) if delimited by two I-nodes, I-C



type (singly connected) if delimited by and I-node and Y- or X-node and C-C type (multiply
connected) if delimited by Y- and X-nodes.

The number of branches and nodes for a given fracture network is strictly related meaning that,
by knowing one of the two elements for the fracture network, it is possible to quantify all its
components. $N_I$, $N_Y$ and $N_X$ can be defined as the number of I-, Y- and X-type nodes and $P_I$, $P_Y$ and
$P_X$ their relative proportions. Once the number of nodes and/or branches making up a fracture array
are known, the connectivity can be visualized using a ternary plot of the component proportions (see
e.g. **Fig. 2**e) or can be quantified by calculating the number of connections existing in the 2D map.
In general, X- and Y-type nodes provide respectively 4 and 3 times more connectivity than I-type
nodes (Nixon, 2013). This forms the basis for creating 2D density maps (see **Fig. 2**d). An array
dominated by I-nodes is isolated, while arrays dominated by Y- and X-type nodes are increasingly
more connected. Connectivity can be quantified by measuring the number of connections per line
$n_{C/L}$ and the number of connections per branch $n_{C/B}$ (see Sanderson & Nixon 2015 for details).

**3   Geological Setting**
3.1   Location and regional structure
The studied siliciclastic strata are Devonian Old Red Sandstone (ORS) of the Orcadian Basin exposed
in the Caithness region, North Scotland. The Orcadian Basin covers a large area of onshore and
offshore northern Scotland forming part of a regionally linked system of basins extending northwards
into western Norway and East Greenland (Seranne, 1992; Duncan and Buxton, 1995) (**Fig. 3**a). The
great majority of the onshore sedimentary rocks of the Orcadian Basin in Caithness belong to the
Middle Devonian and sit unconformably on top of eroded Precambrian (Moine Supergroup)
basement. These sedimentary rocks and the fractures they contain have long been used as an onshore
analogue for parts of the Devonian to Carboniferous Clair Group sequence that hosts the Clair oilfield
west of Shetland (**Fig. 3**a; Allan and Mange-Rajetzki, 1992, Duncan and Buxton, 1995). It should be
noted that strictly speaking, the Clair Group formed in an adjacent basin, in a somewhat different
tectonic setting (Dichiarante, 2017).
Recent fieldwork has shown that the onshore Devonian sedimentary rocks of the Orcadian
basin in Caithness host significant localized zones of fracturing, faulting and some folding on all
scales. Field and microstructural analyses reveal three regionally recognised groups of structures





based on orientation, kinematics and infill (Dichiarante et al. 2016; Dichiarante, 2017). In summary,
these are as follows:
**Group 1 faults** trend mainly N-S and NW-SE and display predominantly sinistral strike-slip to dip-
slip extensional movements. They form the dominant structures in the eastern regions of Caithness
closest to the offshore trace of the Great Glen Fault (GGF) (**Fig. 3**a-b). Deformation bands, gouges
and breccias associated with these faults display little or no mineralization or veining. It is suggested
that these structures are related to Devonian ENE-WSW transtension associated with sinistral shear
along the Great Glen Fault during formation of the Orcadian and proto-West Orkney basins (Wilson
et al., 2010; Dichiarante, 2017).
**Group 2 structures** are closely associated systems of metre- to kilometre-scale N-S trending folds
and thrusts related to a highly heterogeneous regional inversion event recognized locally throughout
Caithness. Once again, fault rocks associated with these structures display little or no mineralization
or veining. Group 2 features are likely due to late Carboniferous – early Permian E-W shortening
related to dextral reactivation of the Great Glen Fault (Coward et al., 1989; Seranne, 1992;
Dichiarante, 2017).
**Group 3 structures** are the dominant fracture sets seen in the main coastal section west of St. John's
Point (SJ in **Fig. 3**b). The comprise dextral oblique NE-SW trending faults and sinistral E-W trending
faults with widespread syn-deformational low temperature hydrothermal carbonate mineralisation (±
base metal sulphides and bitumen) both along faults and in associated mineral veins (Dichiarante et
al., 2016). Hydrocarbons are widespread in fractures in small volumes and are locally sourced from
organic-rich fish beds within the Devonian sequences of the Orcadian Basin (Parnell, 1985; Marshall
et al, 1985). Re-Os model ages of syn-deformational fault-hosted pyrite in Caithness yield Permian
ages (ca. 267 Ma; Dichiarante et al., 2016). This is consistent with the field observation that Group 3
deformation fractures and mineralization are synchronous with the emplacement of ENE-trending
lamprophyre dykes east of Thurso (ca. 268-249 based on K-Ar dating; Baxter and Mitchell, 1984).
Stress inversion of fault slickenline data associated with the carbonate-pyrite-bitumen mineralization
imply NW-SE regional rifting (Dichiarante et al., 2016), an episode also recognized farther west in
the Caledonian basement of Sutherland (Wilson et al., 2010). Thus from St. John's Point to Cape
Wrath (CW in **Fig. 3**b), Permian-age faults are the dominant brittle structures developed along the
north coast of Scotland, forming part of a regional-scale North Coast Transfer Zone translating
extension from the offshore West Orkney Basin westwards into the North Minch Basin (see
Dichiarante et al., 2016).






In the present study, fracture attribute analyses were carried out in areas where Group 3 structures are
dominant, or in locations where there is good field evidence that pre-existing Group 1 faults have
undergone significant later reactivation synchronous with Group 3 age deformation. This approach is
justified based on the fact that the Group 3 structures are the only set widely associated with syn-
faulting mineralization and bitumen and have therefore clearly acted as fluid channelways in the
geological past. There is also good evidence for the preservation of open fractures and vuggy cavities
consistent with these fractures continuing to be good potential fluid-flow pathways at the present day.
No such features are associated with Group 1 or Group 2 structures. Thus we argue that the Group 3
structures are the best direct analogue for the oil-bearing fracture systems that occur in the Clair
Group reservoir in the sub-surface.

**4    Locations and orientation data from the 1D scanlines**
In the present study, 1D scanlines were performed at different scales in the Caithness area resulting
in datasets from regional- (km scale, **Fig. 3**b) and sub-regional- ($10^2$ m to dm scale, **Fig. 3**c),
mesoscale (m to cm, **Fig. 3**d) and micro-scale (μm, **Fig. 3**e).

4.1    Regional- and sub-regional scale
Scanline data have been collected at a regional scale (km-scale) using a tectonic lineament
interpretation map created by Wilson et al. (2010). In their study, the lineament analysis was
conducted at 1:100k scale extending from Lewisian basement outcrops in western Sutherland
eastwards into the Devonian rocks of Caithness (**Fig. 3**b). We performed two scanlines (WTr1 and
WTr2) trending orthogonally to the Brough-Risa Fault, the major N-S trending basin-scale fault in
Caithness (**Fig. 3**b; Dichiarante et al. 2016). Scanline WTr1 intersects mainly NE-SW and NW-SE
trending lineaments, while scanline WTr2 intersects mainly N-S and a few NE-SW trending
lineaments (**Fig. 4**a). Although, datasets with few data points generally give poorly defined
distributions on graphical presentations, it will be shown that the data from these two transects are of
value in the multiscale approach adopted here.

At the sub-regional scale, scanlines have been performed on lineament maps produced from

Google Earth satellite images at 1:1000 scale (pixel resolution c. 10 m). These datasets are limited to





well exposed wave-cut platforms on the coast because the flat topography and thick cover of drift has
obscured the structures inland. The interpreted lineaments from the images were verified during
fieldwork to be faults (large to mesoscale) and joints. The narrow width of the platform limits the
analysis to only one scanline in each locality (DO at Dounreay, SJ at St. John's Point; see **Fig. 3**c).
Fracture strikes and spacing measurements have been corrected using the Terzaghi's Correction (see
dashed red and blue lines in the rose diagrams in **Fig. 4**b-c).

The scanline at Dounreay (DO) is NE-SW trending and intersected mainly NW-SE and NNE-
SSW trending, with a subset of NE-SW lineaments (**Fig. 4**b). The scanline at St. John's Point (SJ)
intercepts mainly ENE-WSW lineaments with subsets of N-S and NW-SE trending features (**Fig. 4**c).

4.2   Mesoscale outcrops
Fracture data along six mesoscale scanlines were collected at three field localities: Brims Ness (BTr1,
BTr2; **Fig. 4**d, e), Castletown (CTr1, CTr2; **Fig. 4**f, g) and Thurso (TTr1, TTr2; **Fig. 4**h, i) where
there is very good exposure. In each outcrop, the position, direction and length of the scanlines were
chosen with reference to the trend of the basin-scale master faults in each area (e.g. ENE-WSW at
Castletown and NNE-SSW at Thurso and Brims Ness; **Fig. 4**d-i). At Castletown and Brims Ness, two
scanlines were carried out to record the full range of fracture orientations: one parallel and one
perpendicular to the master fault set. Scanlines at Thurso differ from the others because they are both
measured parallel and next to a fault zone, resulting in higher values of fracture intensity (see TTr1
and TTr2 in **Tab. 2**). These scanlines are also shorter (< 4m) and record exclusively thin veins. Each
locality is characterized by one (e.g. Thurso) or more fracture sets (e.g. Castletown, Brims Ness).
Where two sets of fractures are present, they generally intersect at high angles to one another and it
was observed that they were active during the same geological event; hence they are analysed here
as single population (Dichiarante, 2017).

Additionally, for each scanline, fracture termination type, kinematics and type of fractures
were recorded (**Tab. 2**). Although fracture terminations are more usefully assessed in a 2D analysis,
we recorded the nature of fracture branch terminations for each structure intersecting the transect line.
These data are reported using a ternary plot (**Fig. 4**j) which shows there is no dominant fracture
termination type. In general, the transects show intermediate to high connectivity, except for scanline
TTr1, which shows a more isolated pattern.





4.3    Microscale scanlines
At microscales, one transect was performed on the oriented thin-section taken from sample SK04
(inset in **Fig. 3**e, left). The fault rock was chosen after fieldwork analysis because of its large thickness
of fault rock and micro-fractured appearance. Field observations also ensured that the age of this fault
was the same as the other Group 3 structures analysed at different scales. The fault rock, from which
the thin-section has been produced, is a typical example of a NE-trending fault with normal dextral
oblique kinematics, filled with carbonate mineralization and red stained (hematite) sandstone-breccia
of inferred Permian age (**Fig. 3**e left, see also **Fig. 5**e). The oriented thin-section was analysed under
an optical microscope and spacing, aperture and the lengths of microfractures recorded. Photo-
micrographs were merged and the scanline was measured orthogonally to the bounding NE-SW
meso-fracture seen in **Fig. 3**e.

**5    1D fracture population results**
5.1    Fracture length - aperture data
MLE distribution fitting and KS tests were performed for all datasets and different types of
distribution (exponential, log-normal, power-law). The recorded range values of trace length and
aperture (or vein width) for each of datasets are shown in **Table 2**. In Table 3 and Table 4 in the
Supplementary Data File, we report the MLE distribution fitting results for both non-truncated
(exponential, log-normal and power-law distributions) and for truncated (power-law distribution)
populations for trace length and aperture, respectively.

Length population datasets yielded values, rounded to the nearest order of magnitude, centred

at ca. $10^1$ m for the sub-regional scale, $10^{-1}$ m at mesoscale and $10^{-4}$ m at microscale (**Fig. 6**a). Aperture
populations are centred between ca. $10^{-3}$ m for the mesoscale dataset and ca. $10^{-5}$ m for the microscale
dataset (**Fig. 6**b).

The plots in **Fig. 6** give an insight into the relationship between cumulative

frequency/intensity (inverse spacing), and length or aperture. For example, at the mesoscale (**Fig. 6**b
centre), the intensity of fractures with > 25 mm aperture is about 0.03 m$^{-1}$ corresponding to a 34 m
spacing. Similarly, the intensity of fractures with > 0.4 mm aperture is between 0.45 m$^{-1}$ and 11.2 m$^{-}$
$^1$ corresponding to 8.9 cm to 2.2 m spacing, respectively. At microscales (**Fig. 6**b left), the intensity



of fractures with > 2.9·10⁻⁵ m aperture is about 155.51 m⁻¹ corresponding to 6 mm spacing whilst the
intensity of fractures with > 3.9·10⁻⁶ m aperture is about 1555 m⁻¹ corresponding to a 0.64 mm spacing.

To examine the possible influence of mechanical stratigraphy on fracture scaling across the

Orcadian Basin in Caithness, we indicate on the fracture size plots, selected sedimentary unit
thickness values reported in previous studies (Fig. **8**a). These include sedimentary laminae thickness
(0.3 mm) at microscale, bedding-range thicknesses of the Lower Stromness Formation (20 cm to 5
m) at mesoscale, and thicknesses of the Ham-Scarfskerry and Latheron Subgroups at sub-regional
scales (data from the in Andrews et al., 2016). Also, the approximate boundary between faults that
can be imagined in seismic reflection images and smaller-scale structures is shown in Fig. **8**a (yellow
arrows) based on well-known empirical displacement-length relationships (a 10m displacement
corresponding to a length of ca 100m following Kim and Sanderson, 2005).

*5.1.1   Analysis of uncertainties: validity of data populations and reliability of best-fit distributions*
In any statistical analysis, the sampled population should be large enough to give a statistically
acceptable representation of the population and to properly determine the distribution type and its
parameters (Bonnet et al., 2001).  The samples sets are statistically valid for most samples after the
first 20 measurements (grey area in **Fig. 7**) because the cumulative fracture intensity of the population
data and its standard deviation (black and green curves, respectively) become reasonably stable. The
uncertainty in the cumulative fracture intensity reduces progressively towards the end of the scanline.

*6    **The scalability of fracture attributes***
6.1    Slope determination –MLE approach
The complete (non-truncated) populations show that a log-normal distribution best describes the data
as they show consistently high percentage fitting values. However, the choice of the best-fit
distribution should not be based on the complete population because the distribution tails
(corresponding to the largest and smallest size fractures) are biased (see also Supplementary Data
File). We therefore also investigated progressively truncated populations in order to validate the
hypothesis. The fitting results for complete log-normal and truncated power-law datasets are
generally similar (see Supplementary Data Files), suggesting that either type of distribution can
successfully describe the size attribute data.




## 6.2   Multiscale analysis

Trace length distribution data from all transects have been normalised using the sample line length
and are displayed together on a single population plot (**Fig. 8**a) which enables us to assess scaling
over 8 orders of magnitudes ($10^{-4}$ to $10^{4}$). The grey region in **Fig. 8**a shows that the multiscale data
can be described by a power-law distribution with overall scaling coefficient close to a slope of -1
centred on a 1 m length fracture with a 1 metre spacing. This power-law distribution implies fractal
or self-similar behaviour of the length parameter over 8 orders of magnitudes which effectively means
that the fracture array maintains the same statistical properties of intensity and length at all scales
assessed here.

The aperture datasets collected in the meso- and micro-scale transects are also shown on a
single population plot (**Fig. 8**b) and show evidence for an overall power-law scaling over 4 orders of
magnitude ($10^{-6}$ to $10^{-2}$) also with a coefficient slope of -1. However, the best-fit line is centred on a
1 mm wide fracture with a 1 metre spacing. This overall slope is indicative of a fractal distribution or
self-similar behaviour of the aperture parameter over 4 orders of magnitude which means that the
fracture array maintains the same relationship between intensity and aperture at all scales assessed
here.

Length attributes for regional faults extend over the estimated thickness of the Devonian rocks
in Caithness by Donovan (1975) as shown in **Fig. 8**a (dashed red line). Although Andrews et al.
(2016) did not report an exact estimate of thickness for the entire Devonian sequence, they suggested
that the thickness reported in Donovan (1975) was overestimated.

## 6.3   Length-Aperture correlations

Trace length and aperture or vein width data are plotted side by side to illustrate the positive
correlation between these attributes over 4 orders of magnitude (**Fig. 8**c). A linear scale length vs.
aperture scatterplot in **Fig. 9**a shows that the data are clustered towards the origin, reflecting the
greater frequency of smaller fractures expected for a power-law distribution (Vermilye and Scholz,
1995). The plot of logarithmic length vs logarithmic aperture in **Fig. 9**b shows two clusters of data
which correspond to the mesoscale population (larger datasets in the centre of the figure) and the
microscale population (bottom left dataset). Small aperture mesoscale data are poorly resolved,



plotting at either 0.01 or 0.05 mm due to the effect of using the thickness comparator in the field. In
the distribution plots, this artefact is removed conventionally by only plotting the highest cumulative
frequency for each aperture value. In contrast, however, in the aperture vs. length plot each individual
data point of the cloud is statistically equally important, although this results in increased uncertainty
at lower aperture values. The logarithmic plot for veins only (triangles in Fig. **9**b) shows a clear
positive power-law correlation between aperture and length, has less pronounced artefacts and
permits an appraisal of the relationship between these two parameters. Line fitting methods suggest
a slope of 0.65 or larger with a $R^2$ of 0.75 (red line in **Fig. 9**b) for all fracture data in this study. A
comparison of veins (triangles) with other fractures including joints (grey dots in Fig. 9b) might
further suggest that veins tend to be shorter for any given aperture.

**7    2D population analysis**
The 2D analysis was conducted at sub-regional scale on a bathymetric map from the near offshore
(**Fig. 10**a) and on the mesoscale using a photograph of a large rock pavement outcrop (**Fig. 10**b) to
provide quantitative assessments of fracture connectivity and self-similarity. The offshore data
provides access to a much larger area compared to onshore, however, the nature of the fractures
themselves can only be constrained by extrapolation from adjacent onshore exposures. We chose to
perform 2D analysis on these areas for two main reasons. First, both contain large numbers of
fractures spread over a large plan view area and therefore were most likely to provide a statistically
meaningful analysis using different 2D methods (e.g. circular scanline windows and box counting).
Second, the difference in size between the two areas gives an insight into to fracture scaling
properties. The fracture interpretation of the bathymetric image enabled analysis of the fracture length
distribution for comparison with the 1-D results, and a topological fracture network analysis of the
fracture nodes.

7.1    2D sampling locations and fracture orientations
*7.1.1    Bathymetry map*
The bathymetry map used for this study is a high-resolution multibeam dataset provided by MeyGen
Ltd (IXsurvey Ltd, 2009) in the area between St. John's Point and Stroma Island where the Devonian
rocks are exposed on the sea floor which has been washed clean by the action of strong water currents
(**Fig. 10**a, raw images in Supplementary Data File). Interpreted faults from the bathymetric data show



ENE-WSW and NNW-SSE orientations. ENE-WSW trending faults dominate in this region (see SJ
rose diagram in **Fig. 4**a) and show "corridor-like" arrays. The orientations of these faults are
comparable to the two main fault sets seen onshore in locations such as St. John's Point (**Fig. 4**c).
NNW-SSE trending faults are regularly spaced (100 to 200 m) in the central part of the area, while
the ENE-WSW trending faults are present across the entire survey. The latter set show two different
spacing values: less than 100 m for the shorter structures and about 1000 m for larger structures.

*7.1.2   Brims Ness pavement photograph*
A similar 2D analysis was carried out using a mesoscale photograph taken at Brims Ness (location in
**Fig. 3**b and raw image in Supplementary Data File). Distortion effects were minimized by analysing
a single photo taken orthogonally to the outcrop pavement and by conducting the analysis in a circular
area to avoid edge distortions. The photo shows three different sets of fractures: N-S, NE-SW and
WNW-ESE trending (**Fig. 10**b). The N-S and NE-SW trending structures form the majority of the
fractures. Most fractures have straight traces and crosscut each other. Three larger WNW-ESE and
NNE- to NE-trending faults were detected. A single curved WNW-ESE trending fault was also
identified (**Fig. 10**b).

7.2    Fracture topology results and fracture connectivity
The bathymetric topology is comprised of 698 I-, 123 Y- and 117 X-nodes, respectively (yellow, cyan
and red squares in **Fig. 10**a) whilst the outcrop topology is composed of 916 I-, 240 Y- and 202 X-
nodes, respectively (yellow, cyan and red squares in **Fig. 10**b).

I-type nodes are regularly distributed in the area while Y- and X-type nodes mainly occur in
the central part of the bathymetry map, where longer ENE-WSW trending faults occur (**Fig. 10**a). X-
and Y-type nodes, which contribute most to connectivity of the 2D system, are mainly localized where
the ENE-WSW trending faults crosscut NNW-SSE trending structures.

The number of connections per line ($n_{C/L}$) and number of connections per branches ($n_{C/B}$)
are respectively 1.18 and 1.1 for the bathymetry image, and 1.53 and 1.22 for the outcrop analysis
(on a scale value between 0 and $\infty$ for $n_{C/L}$ and between 0 and 2 for $n_{C/B}$). This indicates low overall



connectivity for the fracture systems exposed in 2D. The $n_{C/L}$ is also shown in a ternary I-Y-X plot
(inset in the bottom left of **Fig. 10** a and b).

For the bathymetry dataset, the nodal density map shows that the large majority of nodes are
aligned along a series of ENE-WSW trending faults (**Fig. 11**a-b top). The density map shows that Y
and X-nodes are mainly associated with NNW-SSE trending faults and are responsible for producing
most of the connectivity of the system (**Fig. 11**a-b bottom).

7.3    Assessing self-similarity on 2D maps
Circular scanlines were performed to investigate the connectivity of specific smaller areas of the
fracture network on the bathymetry map and mesoscale outcrop photograph (44 and 22 circular
scanlines carried out, respectively – see **Fig. 12**). Circular scanline windows of three different
diameters were used. The numbers of X-, Y- and I-nodes for each scanline are plotted in the ternary
diagrams: blue for small, orange for intermediate and green larger scanlines. The data generally
spread out from the centre of the ternary plot (**Fig. 12**a right and **Fig. 12**b right) and the overall data-
spread is clearly unrelated to the size of the performed scanlines.

Box counting methods were performed in the red-boxed areas shown in **Fig. 12** at the
mesoscale and regional scale to assess whether the self-similarity in the length and intensity attributes
observed in 1D transects are present in fracture patterns in 2D. (**Fig. 12**a-b). The normalized
population plot shown in **Fig. 12**c shows a self-similarity over 1 order of magnitude for both the
bathymetry dataset (**Fig. 12**c, red) and for the mesoscale dataset (**Fig. 12**c, blue). Best-fit exponent
coefficients were obtained using the box counting method plots performed at the two different scales
of analysis: -1.77 for the outcrop photograph and -1.81 for the bathymetry map (**Fig.** c). Both best-fit
curves yielded $R^2$ values of 0.99. The almost identical slopes of ca -1.8 show that the 2D spatial
distribution of fractures sampled at the two different ranges of scale, almost three orders of magnitude
apart, is the same within the resolution of the box-counting method.



## 8   Discussion

### 8.1   Self-similar fault and fracture scaling

Fracture attribute analyses are often conducted on field outcrop analogues because they can provide useful information to bridge the gap between faults imaged in geophysical datasets (e.g. seismic reflection profiles) and fractures observed in borehole data. Our findings show that although individual datasets - particularly at the mesoscale - are best described by a log-normal distribution, this may be the result of sampling bias (incomplete sampling, as discussed in Section 2.1.1). Analysis of truncated populations shows that a power-law distribution can provide an equally representative description of the data. Our results suggest that obtaining an unequivocal power-law fit at a given scale is difficult to achieve because the data may not range over more than 1-2 orders of magnitude. However, when our data were combined from microscale to regional scales, a self-similar (power-law) distribution of fracture aperture and trace length attributes emerges over 4 and 8 orders of magnitudes, respectively (**Fig. 8**c). We suggest that the slope variability observed in individual datasets could result from variation due to local factors such as the presence of damage zones, zones of intense strain or sampling bias. Thus this multi-scale approach can help to reduce the influence of any single dataset and the approach overall can help to reduce uncertainty in assessing the scaling of attributes over a large scale range. If we are correct, then it implies that, at different magnifications (or scales), the dataset structure remains much the same so that the statistical properties can be interpolated to other scales within that range. If present, mechanical stratigraphy at different scales is known to affect the aspect ratio of faults, limiting their vertical size and increasing layer-parallel growth; strata-bound fault distributions are log-normal (e.g. Olson, 2007, see Section 2). Known mechanical stratigraphic boundaries for Devonian rocks in Caithness relative to individual datasets are included in **Fig. 8a** (e.g. cm scale beds at mesoscale), but they do not seem to affect the distribution plots, suggesting that it is not unreasonable to use power-law distributions to describe these data. We note that whilst individual fracture datsets show considerable variability in their slopes (see e.g. **Fig. 6**), over larger scale ranges, data align well along a slope of approximately -1 (see **Fig. 8**) with the same abscissa (intensity) intercept. Previous studies (Odling et al., 1999) on comparable Devonian basins have also demonstrated this type of self-similar scaling, although individual datasets show considerable variability. They investigated fracture length over many orders of magnitudes (1 cm to 1 km) from the Devonian sandstones in the Hornelen Basin (Norway) and showed that, while individual datasets show log-normal distributions, the collective datasets are reasonably well described by a power-law distribution.




We recognise that caution should be applied when using datasets acquired at a given scale to
estimate a fracture attribute on other scales. Censored data might bias the choice of distribution
function that best-fit the data suggesting that log-normal may seem more appropriate even when this
is not the case in reality. However, by extending the scale observation (i.e. by applying a multiscale
approach), we reduce the potential effects of censoring, truncation and variability due to individual
datasets on the overall result and also extend the estimation range for the size parameters such as
length, aperture and intensity. The multiscale approach, together with the analysis of truncated
populations, has enabled us to be more confident in concluding that both single- and multi-scale
populations follow a power-law distribution.

Although, our result remains to be tested with more datasets, the positive correlation we observe
between aperture and length (**Fig. 9**) can provide a basis for a good estimation of frequency and
fracture attributes for large scale (regional) fractures (see next section). The scaling exponent (0.65)
is consistent with recent work that suggests that sub-linear scaling (exponent <1) results when
fractures have grown large enough to be segmented and fracture length increase becomes inhibited
by interactions between segments (see Mayrhofer et al. 2019).

8.2     Applications to offshore fractured reservoirs
*8.2.1   Fracture morphology, apertures and fills*
Most of the fracture apertures measured during the onshore study in the Orcadian Basin in Caithness
are partially to completely filled with either fault rocks, hydrothermal minerals or bitumen; a range
of filling morphologies are preserved (**Fig. 5**a-e). It is reasonable to assume that wholly bitumen-
filled fractures can be viewed as being equivalent to open fractures in a sub-surface reservoir (**Fig.
5**a, b), whilst other veins may be completely filled with minerals/fault rock (lacking bitumen) or
partially filled with hydrocarbon held either in vuggy cavities (**Fig. 5**c), fractured mineral fills (**Fig.
5**d) and/or porous sediment fills (**Fig. 5**e). There are many examples of partly or fully open fractures
in the surface coastal exposures of the Orcadian Basin, but it is difficult to prove whether or not
surface weathering and seawater washing of coastal outcrops has not removed pre-existing fracture
fills. This is supported by the observation that fracture-hosted bitumen fills are most widely preserved
in recently exposed quarry or excavation sites inland (e.g. see Dichiarante et al., 2016).





Dichiarante et al. (2016) presented textural evidence showing that fracture-hosted calcite,
sulphides and oil fills are broadly contemporaneous. Open vugs and fractures are almost certainly
only preserved due to hydrocarbon flooding which shuts down the further precipitation of carbonate
and sulphide in open or partially open fractures/veins (e.g. **Fig. 5**b-e). Observations from the Clair
Field cores by Holdsworth et al. (2019) reveal similar associations between fractures filled, or
partially filled, with similar hydrothermal minerals, younger porous sediment and hydrocarbons. This
suggests that despite differences in source rocks (local Devonian onshore vs. more distant Jurassic
offshore), the Orcadian Basin fracture fills and apertures are a good analogue for the fractured rocks
of the Clair Group.

It is also important to realise that fracture fills of the kind seen in Caithness are not always bad
for the hydrocarbon potential of a fractured reservoir. Wall rock fragments (**Fig. 5**b), early fracture-
hosted hydrothermal minerals (**Fig. 5**c,d), and fills of younger porous sediment all have the ability to
act as natural proppants that hold fractures open in the long term and counteract the tendency for the
present day stress field to close open fracture networks in sub-surface reservoirs (**Fig. 5**a; Holdsworth
et al. 2019, 2020). These fracture fills will however reduce permeability dramatically from the 'cubic
law' relationships of ideal parallel-sided open fractures (Nelson, 1985, Laubach, 2003).

*8.2.2   1D prediction for reservoir volumetrics*
Our approach allows us to provide an illustration of how the fracture scaling relationships established
onshore can be applied as a calibration curve for off-shore reservoirs such as in the Clair Field (**Fig.**
**13**). Published data from Coney at al. (1993) identified three systems of fractures spaced at 30 – 35 m,
100 – 200 m and 1 - 1.5 km hosted in the Clair Group sequences. These scale-ranges are plotted on
the analogue Caithness 1D scaling curves in **Fig. 13**, as a predictor of fracture sizes such as length or
aperture (light grey regions) that might be encountered in a well. Predicted lengths for the spacing
values (inverse of fracture intensity) obtained by Coney at al. (1993) fall in the range of regional to
sub-regional fractures in Caithness, with values of 30 – 60 m length (for fractures with 30 – 35 m
spacing), 100 – 150 m length (for fractures of 100 – 200 m spacing) and 1 – 2 km length (for fractures
with 1 - 1.5 km spacing). Values of aperture can similarly be estimated. Values of 30 – 35 m spacing
have been also measured in our field analogue permitting estimated apertures of about 3 to 3.5 cm.
For larger spacing faults (more than 100 m), values of aperture or fault width can be extrapolated by



extending the slope obtained for smaller scales (light grey area in **Fig. 13**). For example, for faults
spaced 100-200 m and 1-1.5 km, average aperture or fault width is estimated to be 10-20 cm and 1-
1.5m, respectively (light yellow lines in **Fig. 13**). Despite the uncertainties (of 1 to 2 orders of
magnitude) associated with the estimates made here, it is important to note that these larger faults are
likely to only be partially open in the sub-surface (Laubach, 2003). Nevertheless, a 14 cm wide open
fracture was recognized in the core well 208-8 from the Clair Field (Franklin, 2013).

*8.2.3   2D prediction of permeability distribution*
The analysis of 2D datasets using the nodal counting method has shown low connectivity for the
overall systems due to the dominance of I-type nodes compared to Y- and X-type nodes (see ternary
plot in **Fig. 10**a and b). Regions of relatively higher connectivity are localized at the intersection
between larger and smaller structures or in "corridor"-like arrays (**Fig. 11**a-5 and **Fig. 11**b-5).
However, disconnected fractures in 2-D may be connected in 3D and the connectivity density maps
(X- and Y-type nodes) therefore represent a *minimum* estimate of the real 3D connectivity of the
system. A future development of the 2D methodology should be to combine the nodal analysis with
aperture values to produce "weighted" density maps and ternary diagrams. This approach could
provide more realistic values for connectivity flow characteristics of the fracture network.

The increase in connectedness is specific to certain areas of the fracture network (e.g. fracture
"corridors" at sub-regional scale or longer structures at mesoscale). Our findings suggest that large
fractures (faults) will form wide damage zones where there is interaction/intersection between
structures. Correlated with this spatial clustering, we should expect large variability in fluid transport
within the 2D network. If we consider that the spacing between adjacent fracture corridors is about
1 km on the bathymetry map and that from our analogue data the width of similarly spaced corridors
is approximately 10 m, we would predict focused fluid-flow along these structures.

An orientation analysis of fracture intersections has been carried out for onshore faults and
fractures data at St. John's Point (**Fig. 14**a, Dichiarante, 2017), based on its proximity and geological
similarity to the area covered by the bathymetric map which lies immediately offshore (**Fig. 10**a). A
similar plot is also shown for all the faults and fractures data collected in Caithness (**Fig. 14**b,
Dichiarante, 2017). Both datasets show consistent best-fit intersections that are sub-vertical to steeply
plunging to the east, 73/084 and 78/098, respectively (yellow diamonds in the stereonet in **Fig. 14**a-
b). The combination of the connectivity information in plan-view derived from the bathymetry map



and the fracture dip information derived from fieldwork shows that fracture corridor structures and
fracture intersections will be useful in constraining the main fluid-flow direction that should to be
considered when developing the most effective drilling strategy. In general, the calculated steeply
plunging fault/fracture intersections would seem to favour horizontal drilling as opposed to vertical
drill orientations (**Fig. 14**c).

In the analogue bathymetric map dataset (**Fig. 10**a), we observed 1D spacing ranges similar
to those observed by Coney et al. (1993) for the Clair Field. We recognize 100 – 200 m spacing for
NNW-SSE trending faults and less than 100 m and 1 km for ENE-WSW trending faults. The
spacing/intensity values seem to confirm that the Devonian in Caithness is a good analogue for the
Clair Field. Connectivity results from the bathymetry data have shown that these fractures are locally
well-connected in plan-view (**Fig. 11**a-5) and scanline analysis results have shown that these fractures
are potentially permeable with (kinematic) apertures of about $10^{-1}$ m to 10 m producing, in the latter
case, "corridors of partially open fractures" where these are clustered. These localized regions are
believed to provide most of the connectivity of the 2D system and fluid-flow, which is consistent with
the distribution of mineralization observed in the field along corridor-type structures (e.g. the White
Geos Fault locality described by Dichiarante et al., 2016).

Our study shows that a multiscale 1D and 2D data analysis of the Orcadian Basin is a useful
analogue to aid understanding the fracture-dominated fluid-flow patterns in a sub-surface reservoir
(Clair Field). The relationship between aperture and fracture size (e.g. length) is known to have a
major impact on fracture rock permeability (Odling et al., 1999 and references therein). Our
mesoscale description of this relationship together with the multiscale constraint on the 1D fracture
sizes distributions enables us to estimate the kinematic aperture of the largest fractures in the analogue
system even though we have not sampled them directly. The 1D fracture size analysis is extended by
the 2D approach that captures fracture interaction, clustering and connectivity to describe map-scale
spatial variability of the system. These relationships can be directly applied to the Clair Field and
other equivalent sub-surface reservoirs by calibrating the fracture size populations from drill core and
image log data, the spatial properties from seismic attribute data, and the fracture fills from core
description.
A similar methodology may be applied in geological contexts ranging from hydrocarbon
exploration, geothermal reservoir analyses, carbon capture and deep radioactive waste disposal
facilities (e.g. see Pastoriza et al. in review). The straightforward multiscale approach allows direct



comparison between analogues and sub-surface targets and is easy to apply to different areas, dataset-
types and scales to provide important constraints for reservoir modelling and prediction at regional
scales.

**9    Conclusions**
The orientation of fluid conductive faults at basin scale, together with their spacing and connectivity
is crucial to understanding the geometries and fluid-flow characteristics of a sub-surface reservoirs.
Statistical analysis of fracture attributes from suitable outcrop analogues can provide reliable and
robust qualitative (geological) and quantitative (attribute information and scaling) information which
can be used in the design and conditioning of reservoir simulation models.

The Devonian rocks of the Orcadian Basin in Caithness provide a direct analogue for the main
reservoir in the Clair Field and other equivalent offshore fractured reservoirs hosted in similar tight
sandstone strata. The methodology used here represents an alternative to the use of single-scale
datasets in fracture characterization. We advocate an extended approach that integrates datasets
collected at different scales and combines 1D and 2D analysis. The statistical analysis provides a
useful insight into the nature and scalability of the natural fracture networks. Specifically:
-    Our 1D analysis has shown that the population distribution of length and aperture of the
onshore datasets may be represented using a truncated power-law distribution.
-    The multiscale approach shows scale-invariance. The scalability of single dataset can be
extended from 1-2 orders of magnitude (single plots) to up to 4 and 8 orders of magnitudes
(side by side plots) for aperture and trace length, respectively. This illustrates the effectiveness
of the multiscale approach (**Fig. 8**).
-    The positive correlation between vein aperture and length is well represented by a power-law
distribution over 4 orders of magnitude. Although, this remains to be tested with more
microscale datasets, we suggest that this methodology might provide a good estimation of
frequency and fracture attributes for large scale (regional) fractures (**Fig. 9**b).
-    A comparison with published datasets (Devonian Old Red Sandstones in the Hornelen Basin
and seismic data from the North Sea) reveals similar power law -1 slope coefficients to the
one obtained during the present study in Caithness.






An associated topological 2D analysis has provided the following additional insights:
-   The overall connectivity of the 2D system is low and very similar on the two scales of

observation studied (ternary plots **Fig. 10**a and b).

-   However, connectivity is highly variable in the system and appears to be mainly associated

with corridor-like structures (e.g. bathymetry map) at a large scale (**Fig. 11**a-5) and on longer

structures at the mesoscopic outcrop scale (**Fig. 11**b-5). This is particularly important when

considering the fluid transport properties of the system.

-   Box counting methods have shown the self-similarity of fracture analysis over about 1 order

of magnitude for at bathymetry- and outcrop-scales. The datasets have almost identical slopes

showing that the fracture arrays over different scale ranges have the same 2D spatial

distribution (**Fig. 12**c).


Compilations of onshore fracture data show a regional predominance of sub-vertical fault
intersections (3D). This suggest that a horizontal drilling strategy would be favoured were these rocks
to be drilled as a reservoir. The combination of 2D connectivity density maps (plan view) with dip
information derived from onshore structures helps constrain the likely optimal fluid-flow locations
and directions.

Our study demonstrates how a comprehensive and multiscale approach to analogue outcrop
studies may provide a better understanding of the 1D size distribution of fracture networks and map
view variability of fracture networks in an analogue system and how it may be applied to a fractured
reservoir in sub-surface locations.

**Data availability**
Fracture data and results of topological analysis are available at [doi:10.15128/r1cv43nw819](doi:10.15128/r1cv43nw819)

**Author Contributions**



AD designed and conducted the research, interpreted the data and prepared the manuscript. KM
assisted with data analysis and manuscript preparation. RH designed the study and assisted with
manuscript preparation. TB assisted with data analysis, ED assisted with data collection and analysis.

**Acknowledgements**
We are grateful to the Clair Joint Venture Group for funding Anna Dichiarante's PhD project. We
thank Sarah Crammond of MeyGen Ltd for providing the bathymetry data. Riccardo Parviero is
thanked for input on the statistical analysis.

**Supplement**
A supplementary data file containing the statistical method and images used in the analysis is
available at http/:xxxxxxxxxxxx.

**Competing Interests**
The authors declare that they have no conflicts of interest.

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





**FIGURES**
**Fig. 1:** (a) Synthesis of 1D and 2D methodologies to estimate fracture attributes: (i) scanline sampling
(or transect), (ii) window sampling, (iii) circular scanline window and (iv) box counting method
(modified after Zeeb et al., 2013). $L$ = box counting size, $l$ = box size grid. (b) Schematic illustration
of fracture attributes measured along a scanline (or transect). Fractures are represented as straight
lines differently orientated to the scanline AB. Termination types are defined as X for cross-cutting
relationship, Y for abutment of a fracture and I for isolated fracture (or termination against a
lithological boundary). (c) Graphical representation of the Terzaghi's Correction, consisting of
multiplying the scanline spacing $S_i$ by the sine of the angle α between the main trend of the fractures
and the scanline (after Mäkel, 2007).
**Fig. 2:** Population distribution plots for (a) exponential (linear-logarithmic axes), (b) log-normal
(logarithmic-linear axes) and (c) power-law (logarithmic-logarithmic axes) distributions with relative
best-fit equations (top) and sketch of physical meaning (bottom). In the distribution plots, datasets
are shown as black diamonds and typical best-fits are shown as red dashed lines. (d) Examples of
density maps showing higher connectivity where Y- and X-nodes occur, (e) ternary plots showing
that the overall system shown in Fig. d is isolated and (f) self-similarity plot method from **Fig. 1**a
(iv).
**Fig. 3:** (a) Location map of the North Sea with the outline of the Orcadian Basin (light blue area). (b)
Schematic geological map of the North Scotland showing the interpreted fault lineaments by Wilson
et al. (2010) and the trace of the regional scale transects (WTR1 and WTR2) and the location of the
sub-regional transects (DO and SJ). (c) Example of Landsat aerial image showing the trace of the
sub-regional scale transect at Dounreay (DO). (d) Oblique view of the platform at Castletown. The
meter ruler shows the trace of the transect CTr1 (mesoscale). (e) Outcrop photograph of the NE-SW
fault zone where the sample for the thin-section SK04 was collected (yellow star), thin-section
photograph (top) with an example of one of the microphotographs showing one fracture (right). The
trace of the scanline is shown by a continuous red line and a reference line is shown in blue. CW =
Cape Wrath, GGF = Great Glen Fault, fr = fracture, SK = Scarfskerry.
**Fig. 4:** Rose diagrams of fracture orientation data for the transects at (a) regional scale, (b,c) sub-
regional scale and (d - i) mesoscale. Note that the transect strike is corrected for the transects at sub-
regional scale (dashed blue lines in rose diagrams). (j) Ternary plot providing an estimation of the





different type of fracture branches intersecting each transect. N = number of fractures, MAX =
maximum, CI = 95% confidence interval.
**Fig. 5**: (a) Diagram summarizing how the present day aperture of a fracture is related to its
morphology, aperture and fill and the general influence of present day stress. (b-e) Different fracture
aperture and fill types associated with oil in the Orcadian Basin. (b) Photomicrograph of open fissure
with oil fill and wall rock fragments, Thurso Bay foreshore; (c) photomicrograph of partial calcite fill
with vuggy oil fill, Dounreay; (d) photomicrograph of oil-filled brecciated calcite in dilational jog,
Dounreay; (e) Outcrop photo of calcite and red sandstone fill of inferred Permian age, Scarfskerry
foreshore (see Fig 3). All thin sections are taken in plane polarized light, with scale bar = 1mm.
**Fig. 6:** Cumulative distribution plots of (a) fracture and fault trace length for transects at (left)
microscale, (centre) mesoscale and (right) sub-regional scale and (b) aperture and vein width for
transects at (left) microscale, (right) mesoscale. On the plots reported stratigraphic layer thicknesses
are shown as grey boxes. The Ham-Skarfskerry Subgroup (177m) and the Latheron Subgroup (114m)
from Anders et al. (2016) are shown on the sub-regional scale plot, The Lower Stromness Flagstone
(5m) on the sub-regional scale and mesoscale plots. At mesoscale plot the average thickness of beds
(c. 20cm) is also plotted. On the microscale plot, the thickness of individual laminae (c. 0.3mm) is
shown.  Dashed lines and number refer to values discussed in text.
**Fig. 7:** Fracture intensity and standard deviation as function of fracture number for (a) sub-regional
scale, (b) mesoscale and (c) microscale transects. Fracture intensity is unstable for a relatively small
number (< 20) of detected fractures (grey area).
**Fig. 8**: Cumulative frequency plots of (a) fracture length and (b) fracture aperture. (c) Side by side
population distribution plots of length (right side of the plot) and aperture (left side of the plot). Note
that the distance between the datasets in different localities (down to mesoscale) represents the
relationship in terms of order magnitude between aperture and length.
**Fig. 9:** (a) Length vs. aperture scatter plot and (b) Log of length vs. log of aperture for veins (triangles)
and other structures (circle). Linear regression for veins on logarithmic plot is shown (dashed red
line).
**Fig. 10:** (a) 2D analysis of bathymetric data from the area between St. John's Point and Stroma Island
with lineament interpretation and I-, Y- and X-nodes, rose diagrams of lineaments and ternary plot of





node-types proportions. b) 2D analysis of outcrop pavement photograph with lineament interpretation
and I, Y and X nodes and ternary plot of node-types proportions. MAX = maximum density.
**Fig. 11:** Lineament and density maps of nodes for (a) the bathymetry fault network and (b) the fault
network in pavement.   All nodes density map (top) Y, X- type nodes density map allowing a
qualitative assessment of connectivity (bottom).
**Fig. 12:** (a left) 2D topological map of bathymetric data and (b left) 2D topological map of outcrop
pavement photograph showing box counting area and example of performed circular scanlines.
Ternary plots of circular scanlines performed on (a right) bathymetric data and (b right) outcrop scale
photograph. Note in the ternary plot from the bathymetry data the 22 circular scanlines resulted in 16
distinct proportions of I-,Y- and X- nodes. Box counting method applied to (c) bathymetric data and
(d) outcrop scale photograph. (e) Logarithmic-logarithmic scale plot showing the result obtained from
the maps in d and e. Data are normalized by box size and number of fractures.
**Fig. 13:** Sketch of the side by side population distribution plots of fracture lengths and apertures from
Fig. 6. The dark grey areas represent the region where all the aperture (left) and length (right) plots
are localized. Coloured lines represent the distributions at each scale. Orange horizontal lines
represent the reported spacing values for Clair (Coney et al., 1993) and yellow vertical lines represent
the relative estimated aperture values using trends from this study. Note that we extrapolate the
aperture (light grey area) with the slope derived from the microscale and mesoscale datasets.
**Fig. 14:** Lower hemisphere equal area projections of measured offshore data at (a) St. John's Point
and (b) Caithness. (c) Schematic block diagram created by combining offshore 2D density map of
connectivity and onshore dip values. Note that the best-fit of faults and fractures data collected
onshore at St. John's Point (yellow diamond in the top stereonet) is consistent with the best-fit of
fault and fractures data collected in Caithness (yellow diamond in the bottom stereonet, Dichiarante,
2017). MAX = maximum density, MEAN = mean density.
**Table 1:** Basic parameters, definitions and equations provided by 1D and 2D methods (Zeeb et al.,
2013 *modified*)
**Table 2:** Transect data. GPSs = GPS position of the starting point, *N*= total number of sampled
fractures, *J*= joint, *V*= vein, *FnI*= Fracture without infill, *T*= tensile, *Dx*= dextral, *Sn*= sinistral, *I-I* =
"isolated branch", delimited by two I-nodes. *I-Y* and *I-X* = "singly connected" branches, delimited by





one I-node and one Y or X-node. *YY*, *YX* and *XX* = "multiply connected" branches, delimited by two
Y or X-nodes or one Y and one X-node.



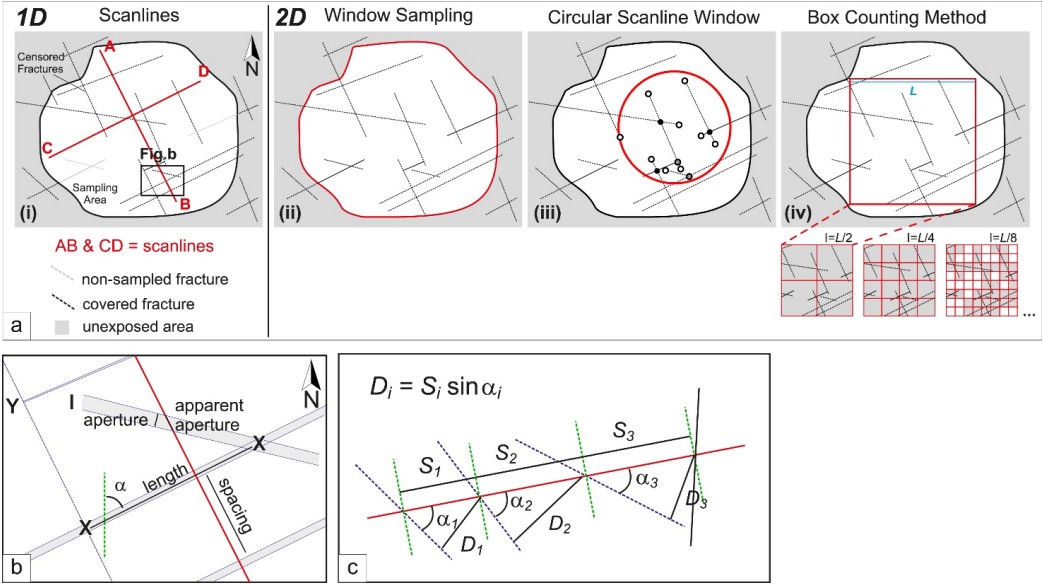

Fig. 1


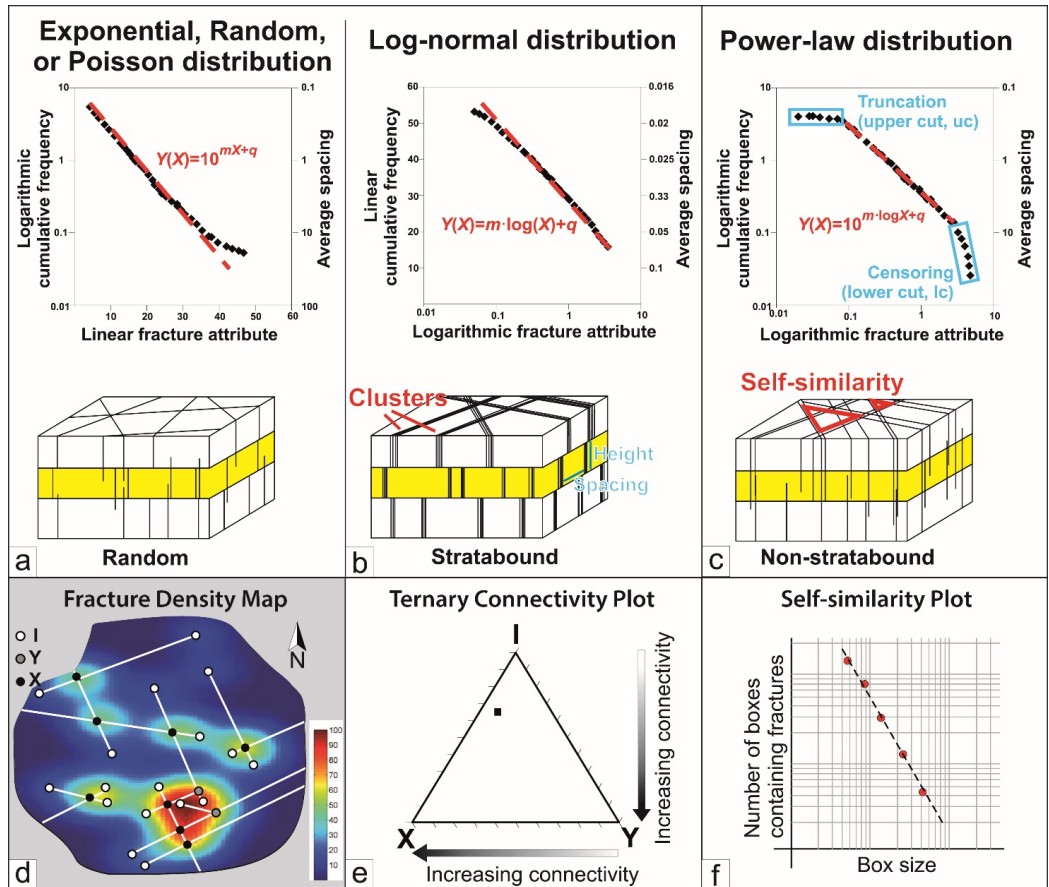

Fig. 2

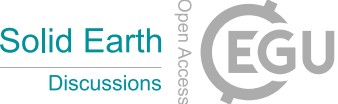

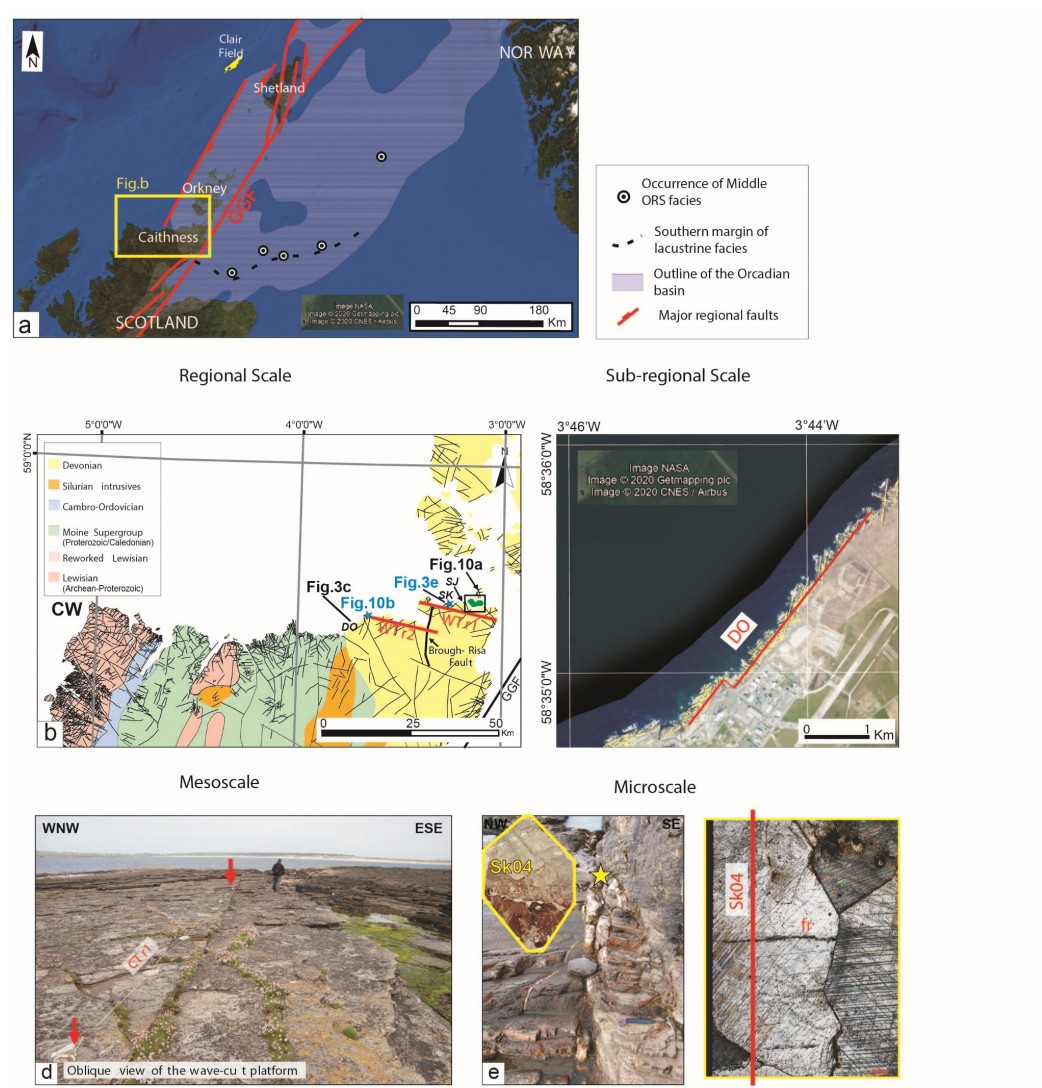

Fig. 3





Fig. 4



Fig. 5





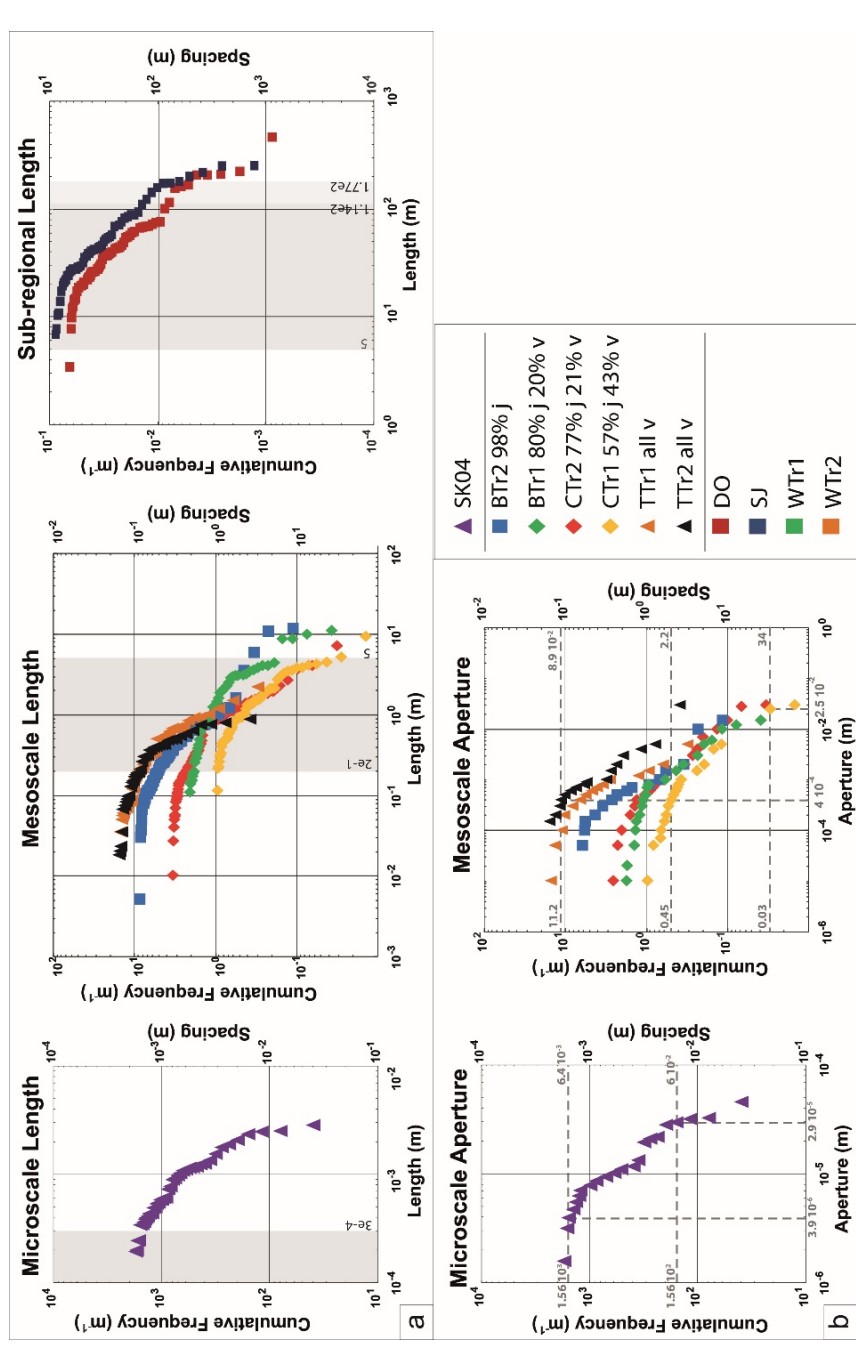

Fig. 6





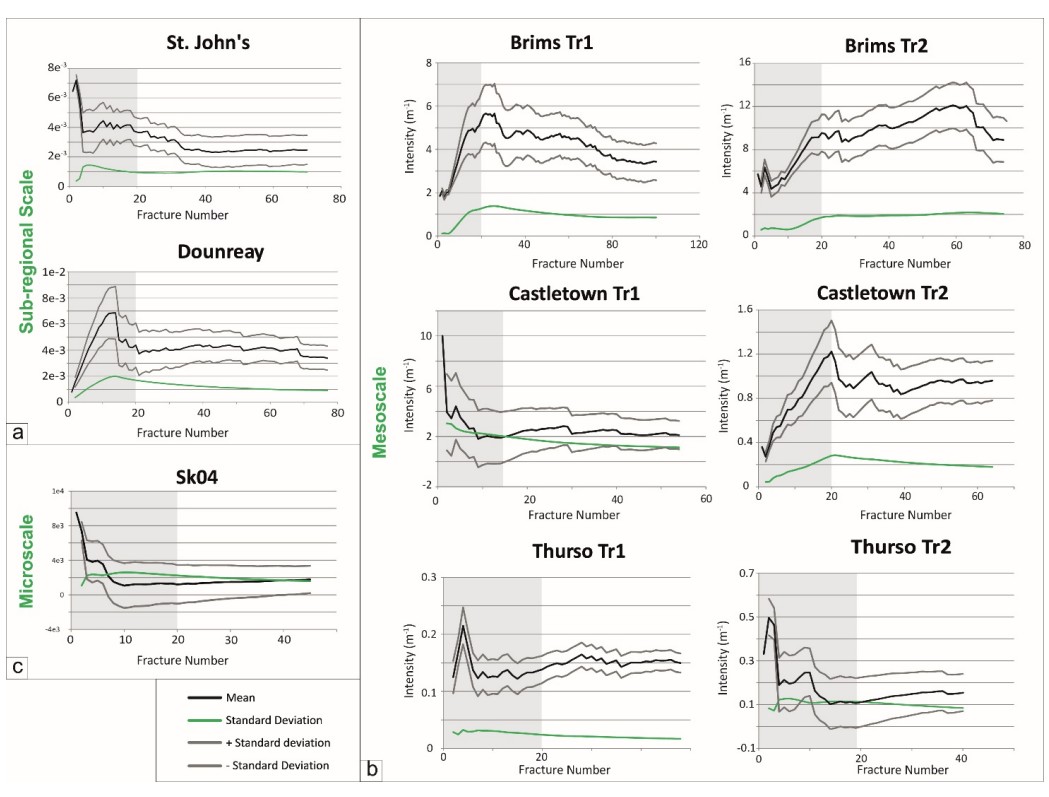

Fig. 7




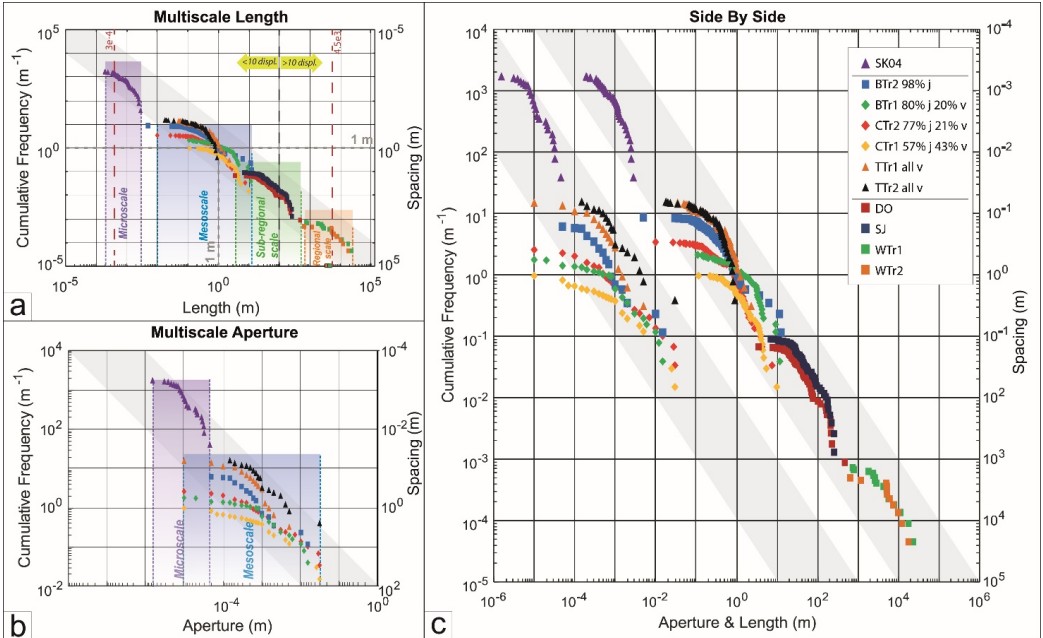

Fig. 8





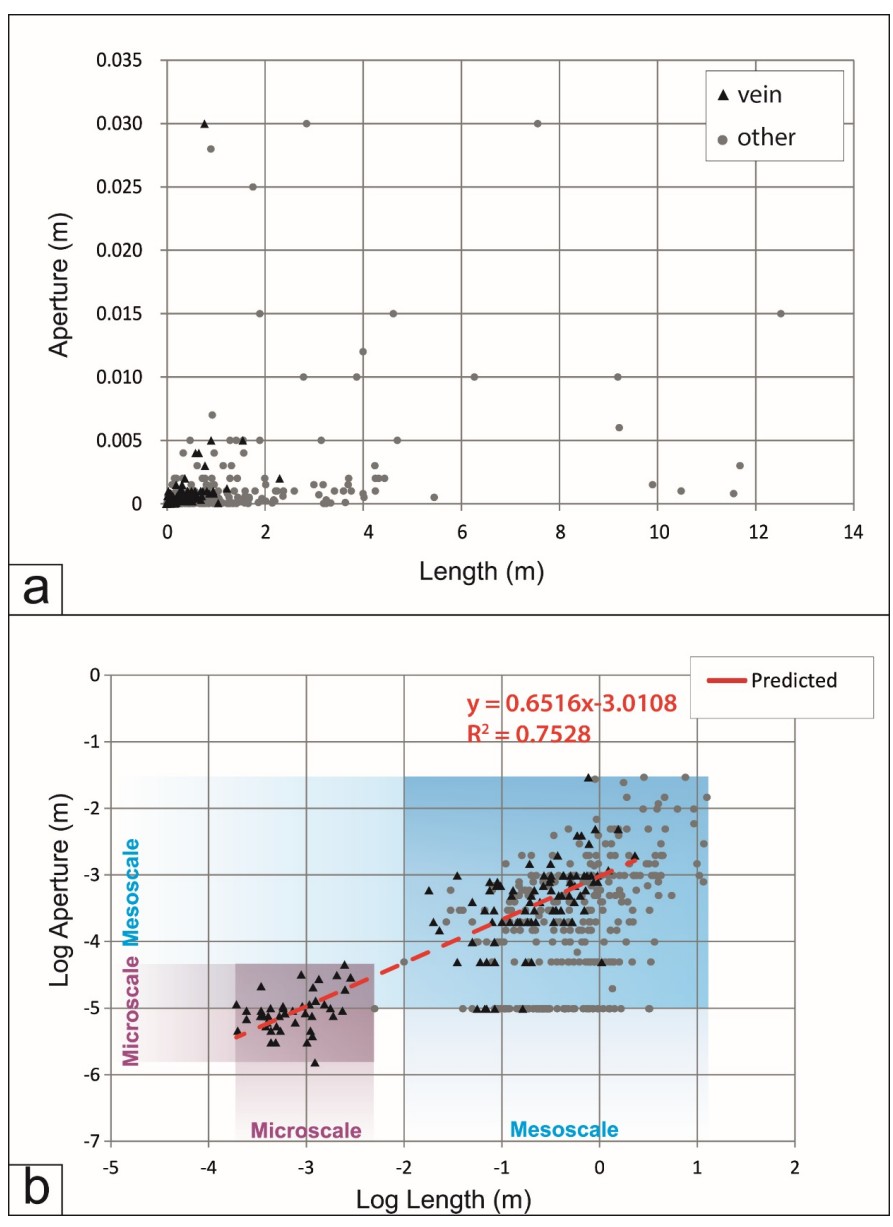

Fig. 9





Fig. 10



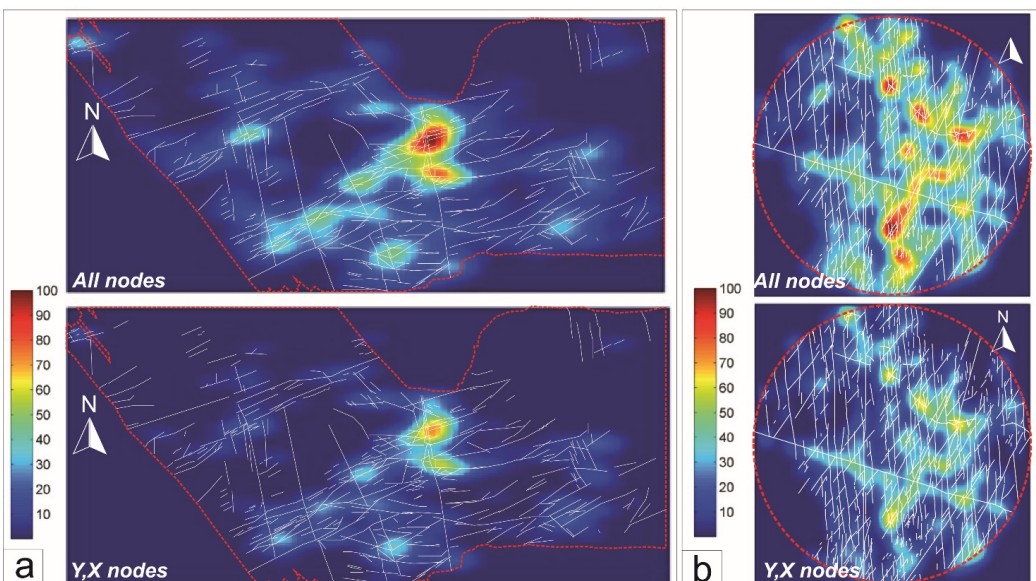

Fig. 11

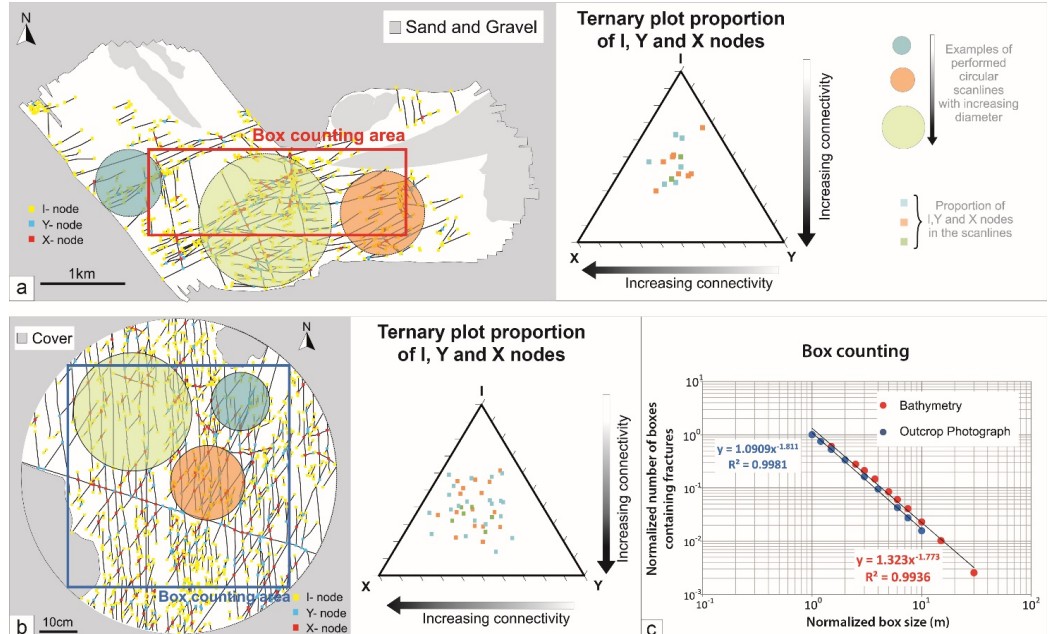

Fig. 12



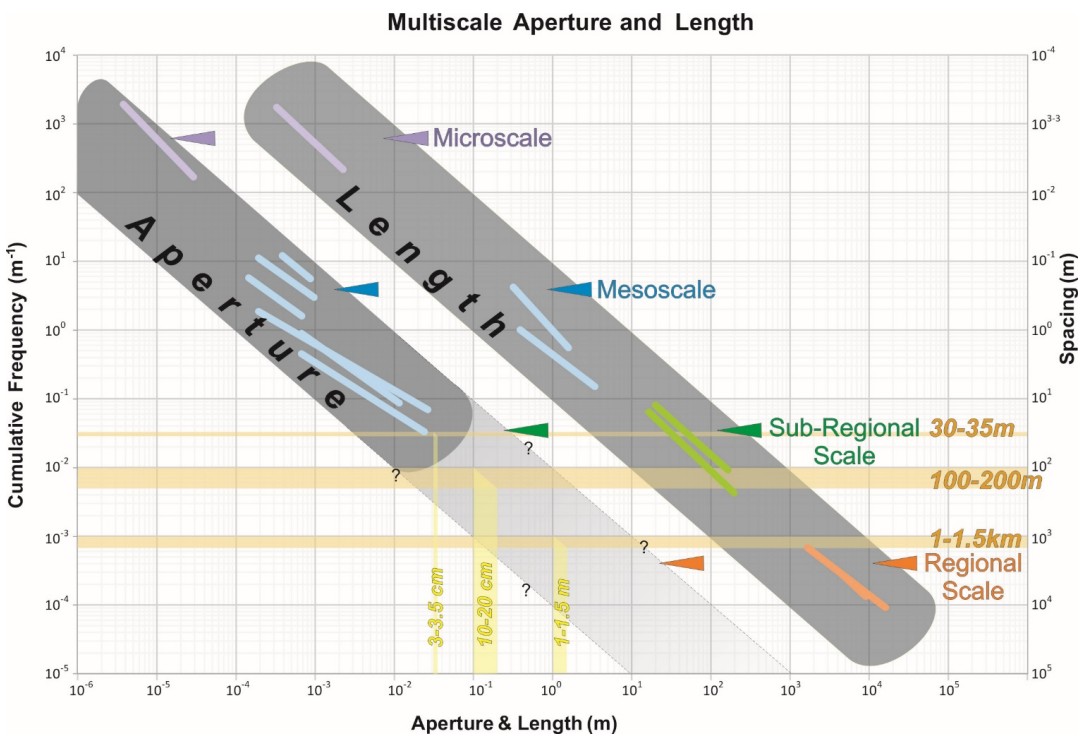

Fig. 13



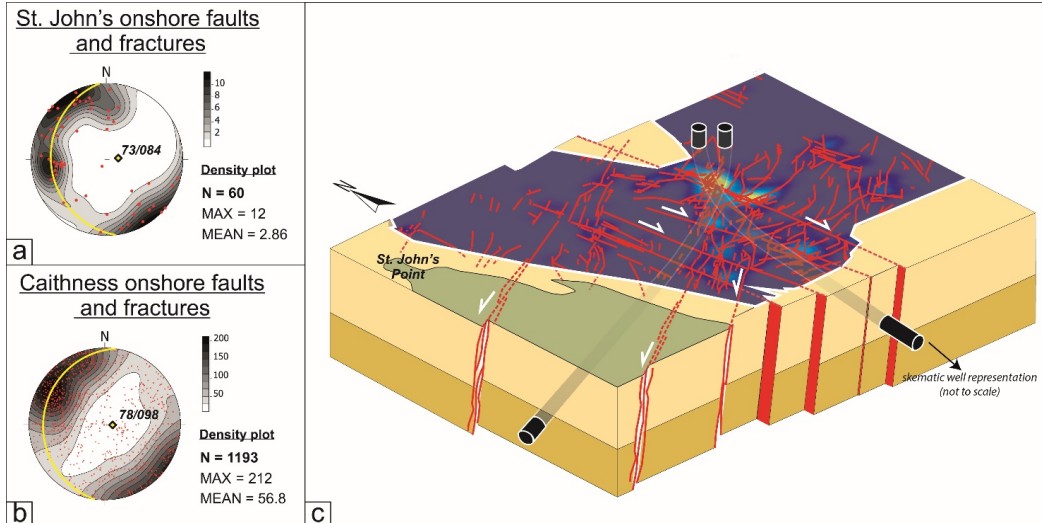

Fig. 14



| Parameter | Definition | Scanline Sampling | Window Sampling | Circular Scanline | Box Counting Method |
|---|---|---|---|---|---|
| Orientation | Orientation of a fracture on a sampling plane (1D) or sampling volume (3D) | YES | YES | - | - |
| Spacing ($s$) | Spacing between consecutive fractures [m] (1D) | $S = l/I$ | - | - | YES |
| Length ($l$) | Length of fracture intersecting the scanline (1D) or sampling area (2D) | YES | YES | - | YES |
| Aperture ($a$) | Aperture of fracture intersecting the scanline (1D) | YES | YES | - | - |
| Intensity or Frequency ($I$) | Number of fractures ($N$) per unit length ($L$) [m$^{-1}$] (1D) | $l = N/L$ | - | - | - |
| Density ($D$) | Number of fractures ($N$) per unit area ($A$) [m$^{-2}$] (2D) | - | $D = N/A$ | $D = m/2\pi r^2$ | YES |



| Name | GPSs | $N$ | Type | | | Kinematic | | | Termination | | | | | Spacing Range [m] | | Length Range [m] | | Aperture Range [m] | |
|---|---|---|---|---|---|---|---|---|---|---|---|---|---|---|---|---|---|---|---|
| | | | $J$ | $V$ | $FnI$ | $T$ | $Dx$ | $Sn$ | $IY$ | $IX$ | $YY$ | $YX$ | $XX$ | From | to | From | to | From | to |
| WTr1 | ND18351 75022 | 16 | - | - | - | - | - | - | - | - | - | - | - | $3.7 \cdot 10^3$ | $3.4 \cdot 10^2$ | $2.3 \cdot 10^4$ | $7.4 \cdot 10^2$ | - | - |
| WTr2 | ND03054 71126 | 11 | - | - | - | - | - | - | - | - | - | - | - | $3.8 \cdot 10^3$ | $1.78 \cdot 10^1$ | $1.8 \cdot 10^4$ | $6.4 \cdot 10^2$ | - | - |
| DO | NC98340 67080 | 76 | - | - | - | - | - | - | - | - | - | - | - | $2.6 \cdot 10^2$ | $0.8$ | $4.8 \cdot 10^2$ | $3.5$ | - | - |
| SJ | ND29312 74823 | 70 | - | - | - | - | - | - | - | - | - | - | - | $1.5 \cdot 10^2$ | $1.2$ | $2.6 \cdot 10^2$ | $7$ | - | - |
| BTr1 | ND04322 71142 | 99 | 80 | 20 | 1 | 94 | 5 | 2 | 21 | 7 | 27 | 19 | 16 | $1.3$ | $4 \cdot 10^{-3}$ | $7.6$ | $10^{-2}$ | $3 \cdot 10^{-2}$ | $1 \cdot 10^{-5}$ |
| BTr2 | ND04360 71157 | 75 | 73 | - | 2 | 75 | - | - | 10 | 11 | 8 | 22 | 21 | $8 \cdot 10^{-1}$ | $2 \cdot 10^{-3}$ | $12$ | $5 \cdot 10^{-3}$ | $1.5 \cdot 10^{-2}$ | $5 \cdot 10^{-5}$ |
| CTr1 | ND18885 69104 | 54 | 31 | 23 | - | 14 | - | - | 10 | 4 | 9 | 4 | 0 | $3.2$ | $5 \cdot 10^{-3}$ | $12$ | $0.1$ | $1.5 \cdot 10^{-2}$ | $1 \cdot 10^{-5}$ |
| CTr2 | ND18922 69088 | 65 | 50 | 14 | 1 | 8 | - | - | 7 | 11 | 17 | 12 | 0 | $4.6$ | $2 \cdot 10^{-2}$ | $9$ | $0.11$ | $3 \cdot 10^{-2}$ | $1 \cdot 10^{-5}$ |
| TTr1 | ND10899 69071 | 48 | - | 48 | - | 48 | - | - | 11 | 0 | 3 | 1 | 0 | $2 \cdot 10^{-1}$ | $3 \cdot 10^{-3}$ | $2.3$ | $3.5 \cdot 10^{-2}$ | $5 \cdot 10^{-3}$ | $1 \cdot 10^{-5}$ |
| TTr2 | ND10914 69036 | 39 | - | 39 | - | 39 | - | - | 13 | 0 | 6 | 8 | 2 | $0.33$ | $5 \cdot 10^{-3}$ | $0.9$ | $1.8 \cdot 10^{-2}$ | $3 \cdot 10^{-2}$ | $1.5 \cdot 10^{-4}$ |
| SK04 | ND26135 74584 | 45 | - | - | - | - | - | - | - | - | - | - | - | $2.2 \cdot 10^{-4}$ | $1.2 \cdot 10^{-6}$ | $2.8 \cdot 10^{-3}$ | $1.9 \cdot 10^{-4}$ | $4.6 \cdot 10^{-5}$ | $1.5 \cdot 10^{-6}$ |