# Peer review of "Fracture attribute scaling and connectivity in the Devonian Orcadian Basin with implications for geologically equivalent sub-surface fractured reservoirs"

_Solid Earth, 2020_

## Short Comment (SC1) · 11 Feb 2020

This is a valuable and very interesting contribution. The study looks at outcrop analogues from the Middle Devonian Sandstone - North of Scotland - for analysing fracture attributes and fracture network connectivity. The authors display a multi-scaled fracture dataset (from km-scale high-res bathymetry maps down to the mm-scale thin sections). Fracture attributes (namely, length and aperture) are collected using a linear scanline method, while fracture network data (i.e. connectivity and fractal behaviour) are collected using a mix of circular scanline and box counting methods. Fracture attributes

are then statistically analysed to extrapolate the best viable underlying theoretical distribution using the Maximum Likelihood Estimation (MLE), jointly with Kolmogorov-Smirnoff for testing the goodness-of-fitness. Fracture network connectivity are instead analysed following the method od Sanderson and Nixon (2015).

My comments are mostly relevant to the statistical analysis presented in this work.

In the current manuscript it is not very clear how fracture length data have been acquired. I suppose that, for the different scale-sizes, the authors have measured the all visible fracture trace length cross-cut by the scanlines. If this is the case I could not find it explicitly mentioned in the manuscript.

I found that the the the 'checkerboard' diagrams (associated to the MLE plots) showing the the variation of the percentage of p-value> 0.05 and percentage of H0 (null hypothesis) with increasing truncated samples are great and practical way of assessing the best upper- and lower-truncation to apply to the sample. Hence, it would be very nice to see these diagram in the main body of the paper rather than in the supplementary materials. In this regards, a possible explanation to the finding that "upper cut has a greater influence over the sample fit". This outcome is a direct result of the current method used in Rizzo et al. (2017a) and Healy et al. (2017) to find the best fitting law (which follows Clauset et al, 2009): this method looks for that minimum value (x-min) for which the theoretical distribution law will hold (either Power-law, Lognormal, or Exponential). Large values therefore, do not influence the fitting. Consequently, the method can effectively account for truncation but not for censoring effects.

I believe that throughout the manuscript you have used erroneously the term 'population'. It is more common that we have access to a subset of the population (i.e., a sample) from which it is then possible to estimate the population parameters using the appropriate sample statistics.

References:

Clauset, A., Shalizi, C. R. and Newman, M. E.: Power-law distributions in empirical data. SIAM review, v. 51, no. 4, p. 661-703, 2009.

Healy, D., Rizzo, R. E., Cornwell, D. G., Farrell, N. J., Watkins, H., Timms, N. E., Gomez- Rivas, E. and Smith, M.: FracPaQ: A MATLABTM toolbox for the quantification of fracture patterns. Journal of Structural Geology, v. 95, p. 1-16, 2017.

Rizzo R. E., Healy, D. and De Siena, L.: Benefits of maximum likelihood estimators for fracture attribute analysis: Implications for permeability and up-scaling. Journal of Structural Geology, v. 95, p. 17-31, 2017a.

---

## Author Comment (AC1) · 18 Feb 2020

We thank Roberto Rizzo for his comments, which we will take into account in our revisions.

With regard to fracture length, we measured the entire visible length of each fracture intersected by the scan-line across 4 scales namely:- thin section (microscale), out-crops (mesoscale), aerial images (sub-regional) and google earth image (regional). The length data for the fractures are affected by censoring where they extended be-

yond the limit of the exposed area. The scale at which the fractures were sampled determined the censoring and truncation effects in each dataset.

We can incorporate the 'checkerboard diagrams' and their explanation in the main body of the text in our revised version as suggested. We agree that these are a useful way to determine the truncation points and help to decide which distribution best describes the data.

We will correct in the revised version any misuse of the term population when it should be sample.

---

## Referee Comment (RC1) · Anonymous Referee #1 · 28 Feb 2020

General comments

The topics of fracture attributes, scaling, and the relevance of outcrop fractures to fractured reservoirs are all of current interest. Systematic measurements of apertures and lengths are valuable contributions to the literature. The paper content is appropriate for the special issue and for the journal. The paper is mostly clearly written and is well illustrated. I believe that the technical content of this MS is for the most part interesting, valid, and defensible. But there are several areas where in my opinion improvement is needed.

[Figure]

The main claim of the paper is that it is possible to combine patterns for all types of fractures (opening-mode fractures and faults) imaged from micro-scale to regional scale to find meaningful size scaling patterns. Another claim is that such broad scale scaling observations can be used by making projections to other scales of interest to get input for or to provide information for 'realistic' reservoir models and as input for fluid flow simulations, etc.

There is room for improvement in how clearly these claims are stated, how they are related to previous work, and how well they are defended and supported.

Specific comments

1. The Introduction should have a clearer statement of claims. The key paragraph from lines 65-82 is mostly an inventory of the approaches used and some comments on curve fitting methods best moved to the discussion. The paragraph should be broken up to separate the inventory part from a revised and augmented section that explicitly spells out the claims; text that could start out 'Here we show that. . .'. A clear statement of the claims is essential. These claims also need to match the Conclusions. Neither of these conditions are currently met. The text doesn't make the claims clear. And the first conclusion (line 640) that the outcropping rocks are a 'direct analogue' is not a conclusion at all. This point was merely asserted in the text without much back up. The comments in line 604-605 seem to point just to a similarity in spacing values. But if this is a major conclusion it needs to be signaled more clearly and the evidence needs to be presented more effectively. It may be easier to just assume that the outcrops may be pretty good analogs and present the evidence for this without making it a major conclusion (but explain what you are doing).

2. The discussion of previous work is not adequate. The account needs to be more complete and more nuanced. Considerable work has been conducted on measuring size scaling of opening-mode fractures and faults and using relationships to project to unsampled (or inadequately sampled) scales for various purposes, including acquiring data that can be used for getting input for flow models. Some of the relevant references are cited, but the scope of this previous work is not clear from the presentation. A more informed and complete account is needed. Also missing are some of findings from previous work that bear on the main claims of this MS. For example, large aperture size distribution data sets for opening-mode fractures have been collected from a wide range of sandstones (for example, Hooker et al. 2014) with the aim of predicting the average spacing/intensity of open fractures in reservoirs, and some of these predictions have been tested with horizontal cores or outcrop analogs. This previous work needs to be accounted for more explicitly. And providing a comparison of the results in the current study to the findings of Hooker et al. (2014) seems like an obvious step for putting the current work into context. It should also be addressed in the paper that extensive size-scaling investigations show that some sandstones do not show wide fracture size scaling ranges (see next item).

3. The limitations of scaling that have been found need to be acknowledged. Some tests like Hooker et al., 2009 show that, of example, microfracture aperture size distributions can be projected over several orders of magnitude to accurately predict intensity at sizes where fractures can impact production. But other studies, for example Laubach et al. 2016, show that in some sandstones, fractures have a narrow (characteristic) aperture size range and accurate projections from populations of small aperture sizes to large are impossible. This raises a concern that is directly related to the claims of the MS, since these observations imply that some fracture patterns do not scale (they don't have scale invariant properties; they can't be projected to or from larger or smaller sizes). What about circumstances where there is evidence of narrow fracture attribute size ranges? The evidence of the literature seems to be saying that some fractures patterns 'scale' but others do not. Taking a for instance from within the area represented by MS figure 3, Laubach et al. 2014 J. Struct. Geol. showed that two adjacent sandstones, influenced by faults or the same population as described in this MS, have drastically differing fracture attributes (size, spacing, porosity preservation). According to the proposition in this MS, these attributes should be predictable by the

MS's regional scaling relation. The test case should be discussed. It's hard to see how the regional could get this right, since these contrasting patterns are on the same scale. But the differences between the two sandstones are just those that would affect reservoir behavior. The MS claims on this topic need to be reconsidered or at least more completely explored in this light.

4. A related problem is in the description of studies that examine fractures having a wide range of sizes. The contrast between 'given scales' versus 'multiscale' is problematic, since 'given scale' seems to imply a narrow size range, but some of the studies cited under 'given scale' cover three or four orders of magnitude in scale. Maybe this is just an oversight. The types of structures analyzed and the size ranges analyzed need to be accurately portrayed. Moreover, since the outcrop structures in the reservoir sandstone analog in this MS seem mainly to be opening-mode fractures, the MS should pay closer attention to the previous work on scaling of opening-mode fractures in sandstone. It's surprising that there is no explicit comparison with the compendium of data in Hooker et al. (2014) for example. Or any discussion of the problems with collecting reproducible length data in sandstones outlined by Ortega and Marrett (which is in the reference list).

5. Is the distinction in this MS between 'given scales' versus 'multiscale' between data sets where the structures are clearly genetically related and of the same type, versus mixed populations of opening-mode fractures and faults that may not be related? The text I think could be read this way although this isn't stated explicitly. This part of the MS may be the most problematic. As noted in the comments below keyed to lines in the text, it is not always clear what kind of structure is being compared or projected. This needs to be corrected. Partly this problem in the text comes from using the general term 'fracture' to mean either opening-mode fracture or fault. This usage is stated right at the outset. But it leads to problems, confusing and obscuring the argument. The case is being made in the MS seemingly that, for example, patterns of faults visible on seismic can be used to predict the size distributions and connectivity of opening-mode fractures at the reservoir/outcrop scale. This is a very considerable claim (I'm dubious). But the claim should at least be made explicitly and defended openly.

6. The claim that multiscale analysis can be useful for informing geological models has been supported by examples from the literature (these should be noted) but the claim that, for example, regional lineaments and seismically detected fault trace patterns can be used to predict meaningful fracture attributes at the grid block or smaller scale seems to me to be a bridge too far. If this is the claim, then a more convincing case is needed to support it. An obvious concern is the projections in figure 8. This figure seems to be saying that aperture and length can be predicted to within two orders of magnitude. What are the error bars on that already really wide prediction? How could such a prediction be used? The authors need to explain how to be useful, 'predictions' can span orders of magnitude (compare the prediction of Hooker et al. 2009 with the two orders of magnitude of size range in the projections of figure 8). Core and outcrop analog data show that fracture patterns at the core and outcrop scale can vary considerably in ways that directly impact fluid flow. As noted above, with adequate samples where microfracture populations are present some of these attributes can be accurately projected over three or four orders of magnitude to predict the attributes of large fractures. But these are cases where the small and large fractures are growing and interacting together in a specific rock type. Contemporaneous, interacting fractures are the ones likely to develop power-law size distributions (Cladouhos, Marrett 1996).

7. The referencing of certain points needs to be made more complete or more accurate. I've flagged instances in the following detailed notes. As it stands now, I don't think the MS properly represents or credits previous work.

8. There are a number of places in the text where reorganization is needed. The Introduction could be clearer. Some of the material in the Discussion looks more like observations/results. I've flagged some of these issues in the detailed line comments that follow. Improving the overall presentation will increase the impact of the paper.

9. I've flagged some areas in the text where meaning is unclear. For the most part the language seems fluent and precise. So, overall substantial conclusions are reached. But in its current form the MS could do a better job supporting the interpretations and conclusions.

Specific comments keyed to lines in the text and technical corrections

Lumping faults and opening-mode fractures together for analysis is a mistake in my opinion. This broad statement about fault and opening-mode fracture size scaling is true to an extent. Marrett et al. (1999) documented power-law scaling across 3 to almost 5 orders of magnitude regardless of rock type or movement mode. This was the study that established that such systematic relations exist and that extrapolation from one scale to some other scale of interest was a feasible approach. It's a surprising omission to leave this paper out.

One thing that Marrett et al. did not do was to mix opening-mode fracture and fault data sets. Doing so requires some defending. It's ok to make the general point that some faults and some opening-mode fracture populations show scaling patterns (although subsequent work shows that some populations do not scale in this way). But it is problematic to lump them all together as 'fractures' if in your description and discussion you let the reader lose track of which kind of structure you are talking about. You are making the claim that it doesn't matter which type of 'fracture' is analyzed—that's fine if you can defend it—but it's not convincing if you just use the all-purpose word 'fracture' in a way that makes it hard for the reader to assess the strength of your claim. For example, in lines (602-612) it's hard to tell which type of structure you mean.

Marrett, R., Ortega, O. J., & Kelsey, C. M. (1999). Extent of power-law scaling for natural fractures in rock. Geology, 27(9), 799-802.

34-37 Consider breaking this initial sentence up into parts. It packs together a lot of claims: a broad definition of 'fractures'; fractures of various types exist over a wide range of sizes; fractures control fluid flow and strength of crustal rocks; fractures influence the behavior of (some) oil and gas reservoirs. That's also a lot of ground to be covered by the three references you call out. But to many readers it may not be apparent to which of the above points each of the three references refers.

One way to revise this would be to cover the most general and less controversial topics about the importance of fractures first, followed by your definitions of geometric and spatial attributes. Then introduce size scaling and spatial arrangement studies systematically (the second clause in your sentence 1). Currently this part of the Introduction seems jumbled.

The line starting 'The heterogeneous distribution...' puts forth claims that seems like they ought to have some support from examples in the literature. And limiting the interest to 'in reservoirs' seems overly restrictive, since many of the examples of real concern for these matters comes from waste disposal, sequestration, and the like. Consider de Dreuzy et al. J. Geophys. Res., 2012. 38 How many readers will pick up on what you mean here by 'scaling parameters'? Maybe move into a more compact paragraph about size scaling.

'Schultz';

By 'in isolation' do you mean in disseminated arrays distant from folds and faults? The opening-mode fractures in such arrays can be closely spaced, and 'isolation' seems like a strange way of depicting that.

It's a dubious proposition that fractures sensu lato as you say can be described with an 'aperture' value. What does this mean for a fault?

And, for fractures at depth in the earth, by their chemical/cement attributes. I think this point ought to be mentioned. As described in a recent Reviews of Geophysics paper, if you just rely on geometric and spatial attributes, one's ability to interpret fractures is seriously restricted, detrimentally impacting the ability to discriminate fracture origins, determine whether or not outcrops are suitable analogs, assess fluid flow and much else. I recommend noting this and calling out the reference where these aspects are explicitly discussed: Laubach, S.E., Lander, R.H., Criscenti, L.J., et al., 2019. The role of chemistry in fracture pattern development and opportunities to advance interpretations of geological materials. Reviews of Geophysics, 57 (3), 1065-1111. doi:10.1029/2019RG000671

Does this reference cover clustering (reviewed in a recent J. Struct. Geol. Special issue)? This list seems to have narrow referencing to cover this wide range of topics. All of these attributes have been treated in depth in the literature prior to the 2015 reference that you cite. And that document is not a review paper.

Also: clarify what you mean by 'continuity'. It's possible to have long (continues) open or sealed fractures, or continuous faults that are both seals and conduits locally. The trace continuity (or connectivity) of lines on an outcrop map or seismic section are no guarantee they represent continuity to flow. Also: see Philip et al. (2005, SPE REE) it is also possible in porous host rocks for discontinuous or disconnected fractures to markedly enhance permeability.

This is misleading. Wellbores and cores provide high-resolution sampling of opening-mode fractures and faults, but the 'fractures' in seismic data are (probably usually) faults. This paragraph gives the impression that the patterns of opening-mode fractures can be discerned on seismic. This has yet to be demonstrated; such fractures are mostly (maybe entirely) below seismic resolution.

You should be explicit about your assumptions. You assume that opening-mode fractures and faults are part of the same population. Presumably if they share scaling patterns if they were growing contemporaneously and interacting (like growth and attachment leading to length scaling, Cladouhos and Marrett 1996). Is there any evidence of this (apart from the scaling data)? The faults and the opening-mode fractures may be genetically unrelated to each other.

One of the challenges with opening-mode fractures is that individually they are

'small' with respect to the attributes that might make them visible on seismic. But they may not be small in other respects. For example, fluid flow. So I think you need to be more careful in this section where you are portraying scales. Because opening-mode fractures commonly have narrow widths and because lengths measured in outcrop are frequently short (in many cases because they are censored by outcrop size) some industry accounts say such fractures are 'small' and can be ignored (Stephenson and Coflin 2015); but some outcrop studies (and some tracer tests) demonstrate that individual opening-mode fractures can be as much as 500 m long (Laubach et al. 2016) and the tracer tests suggest some may be considerably larger (longer). Such long fractures are by no means 'small'. The key to fracture permeability enhancement in a non-permeable host rock is the connected open fracture pathway (e.g., Long and Witherspoon, 1985 JGR; Philip et al. 2005 SPE REE) which is unrelated to visibility on seismic. A long, bed-confined opening-mode fracture might be a more significant feature with respect to fluid flow than a seismically visible fault (the size of the fault may not be the same as its size as a fluid conduit). Philip et al. (2005, SPE REE) showed that very narrow fractures can have pronounced effects on flow.

Stephenson, B. & Coflin, K. 2015. Guidelines for the handling of natural fractures and faults in hydraulically stimulated resource plays. Society of Petroleum Engineers. doi:10.2118/175910-MS.

Laubach, S.E., Fall, A., Copley, L.K., Marrett, R., Wilkins, S., 2016. Fracture porosity creation and persistence in a basement-involved Laramide fold, Upper Cretaceous Frontier Formation, Green River Basin, U.S.A. Geological Magazine 153 (5/6), 887-910. doi:10.1017/S0016756816000157

'networks'?

54-63 I think this paragraph needs some clarification. Is this mostly about how outcrops have been used, or about the extent that analyses have investigated fractures over a wide scale range? The Gomez & Laubach 2006 paper, for example, uses outcrop data to describe fracture aperture size over five orders of magnitude, which seems more than a 'given scale'.

The sentence starts out seemingly about outcrop studies. But the cited reference Makel et al. is a modeling paper; is it really the best call out for the large amount of work that has been done on describing and imaging fractures in outcrop? What about the recent papers by Giovanni Bertotti? See also references in Ukar et al. 2019, Marine & Petroleum Geology, which explicitly covers pitfalls in the uses of outcrop analogs for these purposes and compiles a lot of the relevant literature.

I think you ought to add the clause 'and in horizontal core'. The Hooker et al. 2009 paper focuses on horizontal core. It is also explicitly and example of multi-scale sampling and fracture size analysis.

While I agree that there ought to be more multi-scale samplings of fracture attributes, and it's true that studies of fractures in a narrow scale range are probably more common than ones that look across scales, the way you put it here might make readers think that such studies are rarer than they are. Afterall, the Ortega et al. reference you cite elsewhere is a multi-scale study or fractures and a methods paper on how to conduct such studies and it has 314 citations in google scholar. A review and methods paper that is strangely absent from your list is Marrett, R. 1996. Aggregate properties of fracture populations. Journal of Structural Geology, 18(2-3), 169-178. A more informative accounting here of previous work would be helpful.

Do the references explain why these outcrops are viewed as 'useful analogs'? Or are they about the producing field? I'd prefer to see something more explicit about how you know that these outcrops are valid guides to the specific field in question. Making such a connection is not always straightforward (some common concerns are discussed in Reviews of Geophysics paper mentioned above).

But does your assessment include connectivity to fluid flow? The fractures have cement; cement commonly is more pervasive in narrow segments of fractures. The trace connectivity may well be less than the connectivity to flow. See Olson et al. 2009, AAPG Bulletin.

'a thin section' made from 'samples'; are there more than one of each? Clarify.

'aperture/fracture width parameter'; The MS would read easier if you would define what you mean by the 'parameter' at first use, then stick with it. Are you saying here that you are making no distinction between the 'aperture' of an opening mode fracture and the 'width' of a fault? For 'aperture' do you mean the 'kinematic aperture' in the sense of Marrett et al. 1999? Because many opening-mode fractures in the subsurface are sealed, and so are only apertures in that sense. Even the channels in some fracture cements have finite widths (Landry et al. 2016). Ok: I see you define these, at least for opening-mode fractures in outcrop, on 103. But you need to clarify if you are distinguishing between the 'aperture' of an opening mode fracture and the 'width' of a fault.

The usage here ('whilst...') is awkward. Rephrase. Does 'their' refer to 'fitting methods'?

'fracture attribute' scaling?

Spacing data is pretty uninformative, particularly as seems to commonly be the case if fractures are not regularly spaced.Why not go beyond simple spacing with your scanline data. What about the spatial arrangement (such as implement in the Marrett et al. 2018 J. Struct. Geol. approach)? Also note that application of this method to outcrops and subsurface horizontal well data from the same formation and fracture sets has in some instances found differences between outcrop and the subsurface (Li et al. 2018, J. Struct. Geol.) It's another way to compare outcrops and subsurface.

Did you use the Ortega et al. comparator for width measurements?

Most of the high-quality 'size' data sets that have been published concern 1D aperture size distributions. This is because measuring aperture size on a 1D scanline is pretty unambiguous. 'Length' is another matter entirely. Partly this is due to censoring of long fractures by small outcrops but even knowing what 'length' to measure, with segmented, partly disconnected fractures, is a challenge. Olson (2003) makes the case that the length-aperture relations that reflect fracture growth processes may not be the same as interconnected length, which is what's germane to fluid flow. So I think the generalities in this paragraph need to be treated more carefully. For example, the type of modeling per Olson (2007) (and papers by Michael Welsh) can produce a wide range of types of length distribution in layer-bound systems, and not necessarily log normal. It's also worth noting that the numerous power law aperture size distributions in sandstone described by Hooker et al. (2014, GSA Bulletin) are almost all from layer-bound opening-mode fractures. The 'length scale' defined by layer bound systems for Narr (1991) pertain to the spacing dimension. But fractal clustering (of spacing) has been demonstrated for layer-bound fracture systems (e.g., Marrett et al. 2018, J. Struct. Geol.). So perhaps you should separate generalizations about length, aperture and spacing.

This statement makes it seem like the upper and lower limits are unknowable. But sufficiently complete sampling can discern these limits in some cases.

127-132 The 'state of the art' for fracture aperture size distributions in sandstone goes bit beyond what is portrayed here. Some fracture size distributions in sandstone are wide and can commonly be described using power laws (e.g., Hooker et al., 2014, GSA Bulletin), but other sandstones have narrow aperture size distributions. There are examples in the literature where wide and narrow size distributions occur in adjacent sandstones subject to the same deformation. The differences appear to correlate with rock composition and the inference is that the differences in pattern reflect at least in part diagenetic effects (and so should be more pronounced in sandstones experiencing deformation at depth). So even if it is 'generally accepted' that for many systems power law distributions are useful, in many cases power laws are not an accurate way to describe the fracture population. Wouldn't it make more sense to say that size distributions ought to be careful measured, but there is no reason to think at this point that power-law scaling is the default setting?

What do you mean by 'over several orders of magnitude at a given scale'? The orders of magnitude are of scale.

Are you treating all fractures equally as 'part of the network' or is there some parsing of fractures into sets (i.e., the old fashioned way?) per Hancock (1985). The figures look like they record distinguishable sets.

The 1D power law size distributions could also be said to be 'self-similar'. Likewise, the clustering patterns that come out of some 1D spatial arrangement studies (Li et al. 2018, and other papers on faults in the same JSG special issue). Can you expand on the distinction you are drawing here?

Of connection types?

165-175 It's worth keeping in mind, however, that this is the connectivity of traced lines, not the connectivity for fluid flow.

Is 'strictly speaking' needed?

Moreover, just because the outcrops 'have long been used' as analogs for the reservoir, that does not mean that they are good analogs. A straightforward test suggested by Ukar et al. 2019 M & PG is to compare the progress of sandstone diagenesis in the reservoir target and the potential analog. The burial history (and presumably at least elements of the thermal and loading history) of the reservoirs and the outcrops differ (the comparison is between rocks still buried and those at the surface). The comparison of diagenetic state will at least give you ballpark evidence of how similar the rocks are. Opening-mode fracture arrays are typically low strain features that are sensitive to rock properties so it would be easy for analogs to be 'off' For example, in the western US Cretaceous sandstone outcrop analogs are commonly poor guides to fractures in the same units in nearby basins as documented in core/well log to outcrop comparisons (e.g., Li et al., 2018, J. Struct. Geol.). One of the biggest differences is in how fracture size scaling manifests.

If one of the objectives of this paper is to make the case that these outcrops are good analogs, then perhaps this point could be signaled more clearly in the claims in your Introduction.

Reason for inference of hydrothermal effects?

Have similar features been described from core?

256-261 Sounds interesting. Maybe mention the lengths and numbers of spacings gleaned from these?

This point about the sets being 'active during the same period' is an interpretation. What observations is it based on? Mutually crosscutting relations? If they are a single episode of mutually orthogonal opening-mode fractures, how does that work with your kinematic interpretation? Sounds like biaxial extension. Clarify.

The support for two or three contemporaneous fracture sets (for example, lines 279-280) seems like a key inference, but where is the description of the evidence that these fractures are contemporaneous? The observation that this inference is based on is not mentioned. Since the fractures are said to be partly calcite filled, do you mean mutually crosscutting or mutually abutting relations? The evidence for this relationship should be described, not just asserted.

This sounds like selective sampling. What is the microfracture intensity in the material away from the faults? Such a measurement would be more germane to interpreting the scaling populations of the opening-mode fractures. You would like need multiple contiguous thin sections (like the method described by Gomez and Laubach 2006). Microfractures near the fault does not necessarily mean that there are disseminated microfractures away from the fault.

295-299 This is a pretty short microscanline; cf. Hooker et al. 2009 and 2014 GSA

Bull.

Why report spacing when spatial arrangement (e.g. Marrett et al. 2018) is an option? Intensity is only inverse spacing in a meaningful way if the fractures are not clustered, right?

Normalization like this is a step advocated by Marrett et al. 1999.

Slope of -1?

There is something wrong with this sentence.

Was 'too high'?

380-389 How does this compare with the predictions of Olson (2003)? Do you come back to this?

No comments on what kind of 'fractures' these might be visible in the bathymetry?

Corridor-like arrays, in quotes; what are they supposed to signify. Why not at least cite one of the papers that mentions 'corridors' like Questiaux et al. and/or a recent review of clustering patterns (Laubach et al. 2018, J. Struct. Geol.). J.M. Questiaux, G.D. Couples, N. Ruby Fractured reservoirs with fracture corridors, Geophys. Prospect., 58 (2010), pp. 279-295. With the scanline data you collected it seems like it would be straightforward for you to quantify the degree and type of clustering.

459-463 Do you say what the physical meaning is of the box counting exponent?

465-467 But does the box counting dimension tell you anything about what that spatial arrangement is like? The patterns qualitatively look clustered, locally at least. Do these box dimension mean the patterns are clustered, and by how much? Is it more clustered than random? Can you test this by comparing your results with 1d coefficient of variation or better, a rigorous method like Marrett et al. 2018, J. Struct. Geol.? It seems as though you collected the 1D scanline data that could go into such an analysis so it would be a quick check. You should also at least consider that possibility that box counting is returning artifacts.

So how do the values obtained from box counting relate to the size distributions obtained from the 1D scanlines?

471-500 The text in this section could stand being broken into smaller paragraphs to help lead the reader through the arguments. There are several separate assertions in there. They don't seem well supported.

Is there independent evidence of sampling bias? Or are you just inferring sampling bias because of the mismatch? I don't recall you discussing resolution limits or sample sizes with respect to truncation and censoring. Maybe I missed it. It would help if you did, maybe remind the reader here.

What about the wide range in aperture and length predictions (orders of magnitude; the grey boxes in 8c)?

(fig. 8b) This data ought to be plotted with that of Hooker et al. 2014 compilation, which contains many aperture size data sets, including some from within your general area of interest, and seemingly by the criteria you mention should also match your wide grey bars at least.

Hooker, J.N., Laubach, S.E., and Marrett, R., 2014. A universal power-law scaling exponent for fracture apertures in sandstone. Geological Society of America Bulletin 126(9-10), 1340-1362. doi: 10.1130/B30945.1

Hooker, J.N., Laubach, S.E., Gomez, L., Marrett, R., Eichhubl, P., Diaz-Tushman, K., and Pinzon, E., 2011, Fracture size, frequency, and strain in the Cambrian Eriboll Formation sandstones, NW Scotland. Scottish Journal of Geology, 47/1, 45-56.

What do you mean by 'reduce the influence of an individual data set'? Where do you justify mixing possibly genetically unrelated sets of structures (if that is what you are doing)?

Something is missing here. How does a supposed log-normal distribution of faults relate to the reference Olson 2007?

The 2018 J. Structural Geology special issue on spatial arrangement v. 108, Pages 1-290 (March 2018) has several papers that cover fault size scaling and spatial arrangement. These should be consulted. Also, the meaning of 'bed bounded' in discussing faults needs to be defined in the context you are using it. Obviously at some level of consideration most faults are not bed bounded by definition (at least not in the same way opening-mode fractures can be).

'this type' of self-similar scaling is vague. The specific regression values? Hooker et al. 2014 reports 3822 fractures from 68 scanlines in eight sandstones having 1D power-law exponents of -0.8 plus or minus 0.1. These results need to be engaged with.

I think it's worth mentioning and reminding the reader that this is not the first proposal to measure data at one scale and extrapolate to another. See for example one of the papers you cite where such extrapolations were tested (Hooker et a. 2009). See also Marrett et al. 1999.

Where is your comparison with Olson 2003 (and the discussion and replies to that paper)? The reference is in your list. But you only cite it for very general principles (line 59).

The 'sublinear scaling' inference is due to Olson (2003) and that reference should be cited here.

Why are these basic observations of fracture fills being presented in the Discussion?

'apertures filled with fault rocks'?

Why do you call these 'hydrothermal' minerals? That implies (at least to some) that they were deposited from a hotter fluid moving through cooler rocks (like 'hydrothermal dolomite'). Minerals are common in fractures in sandstone (see the 2019 Reviews of Geophysics paper cited elsewhere for a list of examples). Many of these are not 'hydrothermal' in this sense.

Do you describe these textures?

There is not agreement in the diagenesis community that hydrocarbons necessarily do this. See Bonnell et al. 2006 in Taylor, T. R., Giles, M. R., Hathon, L. A., Diggs, T. N., Braunsdorf, N. R., Birbiglia, G. V., ... & Espejo, I. S. (2010). Sandstone diagenesis and reservoir quality prediction: Models, myths, and reality. AAPG Bulletin, 94(8), 1093-1132.

Indeed. Laubach, S.E., Olson, J.E., and Gale, J.F.W., 2004, Are open fractures necessarily aligned with maximum horizontal stress? Earth & Planetary Science Letters, 222/1, 191-195. This papers describes examples of fractures in reservoirs that open despite reservoir conditions, including stress orientations that should have closed them. Bridges are specifically illustrated. Note however, that as this paper states bridges are not needed to keep such fractures open. All that is needed is diagenetic stiffening of the fracture host rock. The calculation illustrating this is in: Olson, J. E., Laubach, S. E., and Lander, R. L., 2007, Combining diagenesis and mechanics to quantify fracture aperture distributions and fracture pattern permeability: In Lonergan, L., Jolley, R.J., Sanderson, D.J., Rawnsley, K., eds., Fractured Reservoirs, Geological Society of London Special Publication 270, 97-112.

Or reduce flow to none at all.

What you are attempting to do in this section needs to be explained more clearly at the outset. I take it what you are doing is using your scaling data to predict spacing, aperture, and length at a given scale, and testing the efficacy of the prediction by comparing your results to the previously collected spacing data collected by Coney et al. And the length and aperture predictions to some measurements that you made. Is that correct?

. . .us to 'illustrate'

Reconfigure the sentence to make it clearer that Coney et al. is providing the subsurface data. Why 'systems' and not 'sets'? And do you mean that the 'systems' are spaced apart by the values you quote, or that these are typical spacing of subparallel fractures in a set?

560-561 This seems to say you are using spacing to predict aperture.

'more widely spaced faults'?

In what sense to faults have 'aperture'? How is fault 'aperture' related to fault 'width'? It seems like the parameter predicted by the scaling would be something more directly related to fault size, like throw, heave, etc. This needs clarification.

Was this a slant well through a vertical fracture?

By 'core well' do you mean 'cored well'? Was the structure found 'in core' or was it from a cored well, but found on an image log? Your description is ambiguous. Clarification needed.

563-8 This is a section of text where your lack of clear distinctions between opening-mode fractures and faults makes it hard to follow the case you are making. A 14-cm-wide opening-mode fracture is by no means impossible. Excellent outcrop analogs document opening-mode fractures in thick mechanical units of as much as 2 m. And wider calcite-filled veins in reservoir analogs have been described by Hilgers, Urai, and others from Oman. Or is this wide feature part of a fault zone?

Why the quotes around "corridor"-like. Are these corridors, or something that only seems like a corridor but isn't? See also comments in line 582. Most usage of this term seems to follow Questiaux et al. 2010. If you are talking about 'interconnected fracture trace patterns' I suggest you use this phrase instead of the ambiguous term 'corridor' which not only has two very different meanings but also, in the sense that you use it, is not justified. Further to this last point, if what you mean is 'corridor' in the sense interconnected for flow after Manzocchi 2002, you only have information on the connections of the trace patterns, not the flow pathways. Recall the old literature on the Stripa experiment for example.

Questiaux, J.M., Couples, G.D., and Ruby, N., 2010. Fractured reservoirs with fracture corridors. Geophysical Prospecting 58, 279–295.

Manzocchi, T., 2002. The connectivity of two-dimensional networks of spatially correlated fractures. Water Resources Research 38(9), 1162. 10.1029/2000WR000180

Maybe 'minimum' from the point of view of physically connected nodes, but it could still overestimate connectivity to flow. You could maybe estimate how reasonable your numbers are by looking at the reported permeability enhancements at Clair (procedure as in Olson et al. 2009, AAPG Bulletin). With the connectivity you report the enhancement should be huge.

Where is the analysis of spatial clustering? You have the observations for it, but you portray the 'corridors' qualitatively.

From this it seems you are using 'fracture corridors' to mean groups of interconnected fractures. This is confusing usage, especially if it isn't spelled out, since a more common use of 'corridor' is a group of abnormally closely spaced subparallel fractures, i.e., a fracture 'swarm'. These usages are discussed in review in Laubach et al. 2018., J. Struct. Geol. Cluster has been used the way you describe too, and a fairly recent definitions paper quoted that application. But that goes against the bulk of recent usage. In any case, 'corridor' for a tabular feature makes some kind of sense, but 'corridor' for interconnected fault or traces does not. The interconnected traces may not be interconnected for flow (faults or sealed fractures for example) and so may not be 'corridors' for easy fluid flow. I suggest that you spell out what you mean at the outset, then choose something more obviously descriptive as short hand like 'linked traces on fault or fracture map.'

584-587 The expected variability in flow from your networks is a logical jump that doesn't seem well justified. The extent to which the trace connections augment or detract (or do not affect) fluid flow depends on the character of the element. Some faults may be sealing; some numerous and interconnected opening-mode fractures could be sealed. Clusters of 'features' around some faults may be deformation bands that could impede or not affect flow. The flow (if there is any) also depends on the head, if any. This section needs further thought.

589-595 Should this material be in 'Results'?

The topics seem to jump around. This text covers spacing (again). Maybe combine all the information for each attribute info in one place.

The faults are well connected?

602-612 This paragraph is confusing. You seem to be talking about both 'faults' and 'opening-mode fractures' and 'corridors' (some undefined level of clustering). Or do you mean opening-mode fractures associated with faults?

Opening displacements of 10 m? Any bit drop data from Clair to confirm this?

607-608 How do the scanline results show this? Maybe just confusingly put.

614-625 The structure of this paragraphs needs work. It is a mixture of inferences and assertions, but they don't seem to flow one from another.

Your discussion or aperture, length, and connectivity should be more nuanced.

'aperture' is one element of 'size', so this sentence seems awkward. Also, Philip et al. 2005 SPE REE showed that for fractures in slightly porous rocks length is what matters; aperture size is irrelevant. Cf Long and Witherspoon 1985 on connectivity.

The call out to Odling et al. doesn't seem to fit what you are saying here, which sounds a lot like the parallel plate model. The aperture effect (cubic law) in any case needs to be modified considerably when discussing flow in rocks that have finite host rock permeability. Long open fractures can produce considerable permeability enhancement even if completely non interconnected (Philip et al. 2005, SPE REE); and as Philip et al. showed, in those case aperture doesn't matter. Since the sandstone in Clair are porous and permeable apart from the fractures, this is the circumstance likely to apply there.

614-615 The claim of 'usefulness' is I think different from what you've shown. If your claims have been proven, you have shown that your outcrop analogs provide a reliable and perhaps even quantitative view of fracture attributes in Clair field that are otherwise very challenging to measure. Some of these attributes are ones used in reservoir simulation and decision making. If the results are valuable to decision makers, that will depend on the decision making process, the costs to acquire data, whether or not behavior will be changed, etc. I'm just suggesting that you choose your words carefully. Almansour et al., 2020, SPE Re. Eval. Eng. doi: 10.2118/198906-PA has an example of assessing the value of fracture information in an economic/decision making context.

The mixing of terms (fault, fracture/opening-mode fracture) and your scaling analysis leads to a confusing claim here. The 'largest fractures' are faults in your accounting, but how do we know they necessarily have any opening displacement associated with them? Some faults are tight and lack opening displacements, and it would certainly be unusual for a fault to have a large opening displacement along its entire length. This needs clarifying.

Reference not in list.

634-638 This text sounds more like Introduction. How is it a conclusion of this study?

640-641 This statement about the outcrop being a good analog is framed as a conclusion, but the text seems to merely assert that the outcrops are good analogs. There are some observations about fracture petrology and fracture patterns in the outcrop and in the subsurface, but it didn't seem to me that you built a case for such a definite conclusion. Did you alert readers in your Introduction that this is a claim you are going to make? Or do you want to tone this down and say "based on evidence x, y, and z, the Devonian rocks of the Orcadian Basin in Caithness are plausible analogs for the main reservoir, etc. Based on that inference, we. . .'

Isn't this the first use of 'vein aperture'? Despite it being a widespread old term, I don't see any value in retaining it for use in sedimentary rocks. See the short discussion of the term in the 2104 Gale et al. AAPG Bulletin review of fractures in shales. Why not just all these opening-mode fractures and specify the mineral content?

Introducing conclusions about drilling strategy in the Conclusions? Did I miss the discussion of this topic?

The very broad range of aperture and length predictions, for example in figure 8, ought to be discussed where you make claims and conclusions about how useful your findings are for practical application. Are aperture and length predictions that are within two orders of magnitude likely to be practically useful? How can you demonstrate this?

And why not compare to the claims and the uncertainty ranges in the Hooker et al. 2009 across-scales predictions?

Figure captions

Is the box counting method detecting artifacts?

876-878 Is this much information on the standard Terzaghi correction really needed?

Is this real data, or are these example distributions?

To make these figure captions more stand alone and clear, the 'where' and 'what' information on the various scales (a-i) and orientation patterns should be stated in the figure caption. Are these observations all from the target sandstone?

'general influence of present day stress' is vague.

'aperture and vein width'?

Figures

The figures numbers and quality are good. As noted above it would be advantageous to graphically compare results with those of Hooker et al. 2014.

Figure 2. The height categories Random, Strata bound, and Non-stratabound don't cover the most common subsurface fracture height pattern as documented in fractured sandstone core; that height pattern was called 'top bounded' by Hooker et al. (2013, J. Struct. Geol.). The classification proposed by Hooker et al. (2013) has proven to be useful; it's also replicated in Gale et al. (2014) shale fracture review paper. I recommend that you apply it.

Fig. 3, 'f' lacks a label; both 'e' and 'f' need graphic bar scales. The inset in 'e' and 'f' are both quite small and hard to read. From the figure caption it's hard to tell what is meant to be portrayed in e and f. The photomicrograph appears to show blocky, twinned calcite (is this inside a vein?) possibly containing a fracture ('fr').

Fig. 6 Why not graph the micro and macro data on the same plot? Ok; I see you have this in figure 8. How do the aperture size distributions compare with the values reported by Hooker et al. (2009; 2014)? These figures out to be arranged such that the slopes are not distorted by having different scales.

Fig. 8 The aperture size data should at least in the text be compared and contrasted with the compilations in Hooker et al. (2009; 2014).

Fig. 9 The significant figures on the regression look too high. Check.

---

## Short Comment (SC2) · 10 Mar 2020

**Billy Andrews**

billy.andrews@strath.ac.uk

Received and published: 10 March 2020

Title: Fracture attribute scaling and connectivity in the Devonian Orcadian Basin with implications for geologically equivalent sub-surface fractured reservoirs

This paper provides an extensive dataset investigating fracture properties over several ordered of magnitude using 1D and 2D approaches. The authors provide evidence for power-law behavior over 8 orders of magnitude for fractures (faults, joints, and veins) in the Middle Devonian sandstones of Northern Scotland. The study represents one

of the few multi-scale approaches to fracture characterization, and as such is of clear importance and relevance to the readers of Solid Earth. However, I have a few concerns that I would like to discuss, and in support of the open review system a number of minor points throughout the manuscript.

Major comments

1. The analysis of several sets and fracture types (fault, vein, and joints) within the same population.

Overall, I strongly agree that there needs to be more multi-scale analysis into fault and fracture attributes, and so far many researchers have remained in there 'safe space' (e.g. field geologist rarely assessing more than an outcrop scale). However, the underlying processes behind the formation of faults and joints are completely different, the mineralization can occur at several times during the structural evolution under different stress states. Additionally the underlying controls has been shown to change through time. For example, pre-existing structures can act to limit fault/joint propagation (e.g. Andrews et al., in review; Wilkins et al., 2001), and structural diagenesis change the nature of the mechanical stratigraphy (e.g. Laubach et al., 2010).

You provide a comprehensive structural evolution of the basin in section 3.1 and split the structures into three groups. While you state that you predominantly focus on G3 structure (L226-227), you also look at where G1 faults have been reactivated. Have these structures been assessed separately from the rest of the trend? It would be interesting to know whether the reactivation either caused reactivated G1 structures to have larger or shorter trace lengths. The reversal of the stress states from G1 to G3 will also likely to have caused the propping open of reactivated faults from G1. Has this had any effect on the aperture of these features.

With this in mind, along with the clear age relationships presented in the paper I find it odd that the fractures are not considered as sets. I think this would lead to some sub-trends falling out of the data and explain some of the plots that display trend changed. I

SED
wonder about the viability of using 'kinematic aperture' when investigating veins, particularly when comparing this aperture with joints whereby the aperture could have been significantly altered by later tectonic events. I would also suggest that the kinematic aperture will always plot higher than you would expect for a given length due to the propping open of voids, or multiple crack-seal events.

2. Subjective bias during the digitization of fracture traces, with particular reference to the 2D analysis.

My key concern regarding the digitization, and later analysis, of the fracture networks relates to the 2D analysis presented in Figure 10-11. In a number of places (see at-tached figure for rough outlines for the circular window) it appears as if 'weathering' boundaries, or indeed the ruler for the circular window has influenced topological sampling. Several i-nodes may be observed both along the boundary of the sand and gravel to the east of Figure 10a, and along the N-S trend of the ruler. This pattern is also observed in the density plots presented in Figure 10b. A less stark, but still clear effect, is the presence of weathered sections to the NE and SW of 10b.

I imagine that this is due to feature not being interpreted under areas of no-exposure, something that was shown to greatly increase the level of subjective bias during the collection of fracture data (Andrews et al., 2019). The knock on effect on the topology will be to introduce more i-nodes into the system and hence decrease the estimate of connectivity. If a directional connectivity is then considered, the effect of the ruler would produce a artifact and 'barrier' to flow in the NS direction when there are in fact many fractures that join the NS and NE-SW sets.

3. Power-law vs log-normal

I found this one of the most fascinating things about this manuscript and felt it raised some very interesting points of discussion. Overall, I am not sure which way to fall on this argument and I am certain that a power-law would do a poor job of describing my data from the Carboniferous (coal measures). For units which display little mechanical
stratigraphy, or the mechanical stratigraphy has been 'lost' prior to deformation then I can see how power-law relationships could be very powerful. I wonder how many of the scale dependent mechanical boundaries (e.g. bedding vs facies) are hidden in the scatter of power-law relationships, particularly as it is often difficult to sample multiple scales of either at a single site. Therefore, could using he 'scatter' in a power-law relationship help us input some useful geological data into geo-models even if the subtleties are lost? Only similar studies in different lithologies and tectonics settings can hope answer my questions. How much scatter is 'ok' to produce a good enough geo-model? Although I believe there are several points I don't agree with, particularly in the clumping of 'fractures' and the way in which some of the 2D data was digitized, the manuscript has presented an interesting dataset and judging by the discussion forum began an interesting debate. Overall, I think given a greater discussion of limitations this manuscript will make a useful contribution to the field and I look forward to reading the final version.

Line by line comments

L34-37: As an opening sentence I feel this opens the manuscript up to criticism, in particular clumping the faults, joints, and veins. As mentioned in my major comment 1, I have a fundamental issue with combining these datasets, and I think many others will question it too.

L56: What do you mean by '3D', do you mean truly 3D using such methods as xCT or 2.5D using outcrop photogrammetry or LiDAR?

L61-63: Strongly agree with the need for this work, however, I feel too many sub-sets have been clumped together, see major comment 1.

L71-71: In assessing the connectivity of a network is it worth considering the effect of the chance in scale. Several authors have pointed out that the comparison on connectivity over many scales can lead to issues, and was one of the key drivers for Olsen et al., I believe in there 2004 paper, to assess aperture instead of trace length or connec-

SED
tivity. Subtle chances in observation also lead to differences in interpretation (Andrews et al., 2019; Peacock et al., 2019), with higher scatter observed at higher resolutions (Scheiber et al., 2015). Has the role of subjective bias on this dataset has been considered?

L79-82: I feel this sentence needs to acknowledge the limitation, otherwise it has the potential to lead to authors attempting to 'read-off' your scaling relationships without considering the geological implications.

L89: change 'sizes' to trace length. Also is all trace length bed-parallel? Or does the dip of bedding vary across the section? Do the authors have any inclination of the aspect ratio of faults, joints, and veins, and hence any controls of mechanical layering.

L98: with fracture orientation collected, and a good handle on the geological evolution, I am surprised that no analysis of sets was considered. If it made no difference to the scaling relationships then this is an important observation, however, I would be surprised!

L98-99: I think the textual and fill aspect of this paper is a nice aspect of the multi-scale analysis.

L101: With a number of the features being filled, I wonder the applicability of using the connectivity of the while network? If we are looking to investigate flow surely it is best to only include those features which are conducive to flow, unless you feel features can be 'stimulated' for improved recovery.

L103: 'Kinematic aperture' – I have concerns on the applicability of using this for veins, particularly for veins that display multiple crack-seal events. This will always cause your feature to display an aperture higher than expected for a given trace length, as the feature will have been reactivated along, or through, the previous mineralised fill. Due to aperture being one of the key controls on DFN modelling, should we instead not be using the effective aperture?

SED
L119-121: Log normal distributions can also be caused by pre-existing structures influencing the growth of later structures.

L134: I feel with any multi-scale analysis like this the effect of scale on the interpretation of features is important, and such the role of 'subjective bias' should be considered. A slew of papers have recently came out on the subject including Andrews et al., 2019; Peacock et al., 2019; Scheiber et al., 2015; Shipton et al., 2019; Gibson et al., 2019). What scale we make our observations has a fundamental effect on how we interpret a network.

L139: Does a power-law relationship fit all orientations?

L141-143: I think this is a KEY point of the paper and in my opinion the strongest part and as such should not be hidden away in the supplementary information. It provides a very useful practical example of how the fundamental work outlined by Healy et al., 2017 and Rizzo et al., 2017 can be applied to an extensive dataset. I think figures S2 and S3 should be incorporated into the main text, if you felt that would be too many figures then I don't think all examples from figure 7 need to be included, and 1-2 could be included as a combined figure with S2 and S3.

L155: What was a) the scale of resolution of all maps/photos and b) the 'digitisation' scale (see Major point 2).

L169: state why plotting it in a ternary diagram is useful, e.g. Mauldron et al., 2001.

L191: With clustering discussed, I wonder how you look at your connectivity and how this varies across the mapped areas at different scales. How does this effect other fracture attributes (i.e. in the high intensity zones is the trace length lower? In your smaller scale analysis how many samples do you think are a) away from faults and representative the 'background', b) close to faults and potentially representing a small but important sub-set of your dataset, and c) missing faults due to coastal erosion etc. obscuring areas of high fracture intensity (i.e. exposure bias, Shipton et al., 2019).

SED
L233-235: you state that G3 structures represent the best analogue for the Clare field, however, on line 188 you state the areas formed under different tectonic settings. I wonder if the authors could comment on this?

L255: what scale were the lineaments mapped at?

L268-270: It would be useful to include the length of the scanline in the main text here to remind the reader on the scale of observation.

L272-273: was there any difference in fracture properties between the orthogonal scanlines?

L278-280: Interesting point, it's rare to see two sets that share fracture properties and I assume they formed where sigma 1 and 2 where very similar? However, I would still like to see the trace length distributions split by orientation as this would back up your interpretation that they can be classified under the same population one set didn't affect the others growth.

L282: was it just Ni, Ny, and Nx that was connected, or whether it terminated into a stratigraphic layer?

L286-287: consider including the percentage of connected branches here, what is 'intermediate and 'high' connectivity?

L290: Although I like the use of microstructures in this way, I do wonder how representative the authors feel a sample taken within a fault zone is? Particularly with the local rotation of stress fields commonly observed here.

L297: was spacing measured orthogonal for all sets? Slightly confused on how spacing was extracted here. Was the Terzaghi correction undertaken for all features aligned obliquely to the scanline, and was spacing considered by sets?

L303: MLEs are the 1st results you present.. another reason why the supplementary information should be incorporated into the main text.
L309-312: I think this paragraph is unnecessary and misleading (your only centred due to censoring issues) and I think you should just reference the figure.

L322-325: This is a very interesting point and raises questions on what controls the mechanics of the rock mass at the time of faulting/jointing/veining. Another interpretation is that the scale at which these occur is variable, so the clear change you observe in the seminal mechanically layered sequence work is 'watered' down by a variability in the thickness of the layer. I was not totally convinced from the data that it had no effect, and also for the wave-cut platform examples it appears that the digitised network was on the bedding plane. This means that the mechanical stratigraphy will not have been encountered in that plane of observation.

L364: I find the fact you see this relationship over 4 orders of magnitude very interesting as I would expect the processes to change. Your data therefore suggests that this chance is either hidden in the scatter or that there rocks display a continuous set of processes across these scales.

L367-370: the point of this paragraph is unclear, consider rewording or removing

L389: is this relationship in veins not due to the use of 'kinematic aperture'? with vein fill either 'propping up' the void space during periods of high fluid pressure such that aperture is not reduced with further tectonism, or where multiple crack-seal events occur the 'effective' aperture at the time of fluid flow is along, or cutting, the previous fill.

L396-L397: How does bedding vary across the section? How many structure cut across stratigraphic layering?

L429-431: The different number of nodes in each area makes this difficult to compare in the text, consider changing to a ratio.

L436: you state that ENE trending structures cross cut NNW trending structures, does this have no effect on your fracture attributes?

SED
L439: How much of this change in connectivity do you think is due to the chance in scale of observation? Nixon et al., 2012 provide a nice discussion on the effect of scale on connectivity.

L444-445: what is the fill of the ENE trending faults, and how will this effect the connectivity of the system? Does this cause an increase in fracture intensity, or is it only that the earlier fractures are offset by the later structures such that the number of nodes increases?

L465-467: Very interesting point!

Section 8.1, Paragraph 1: Please split this paragraph! It becomes unfocused contains several distinct points. Potentially on Line 483?

L473: I think the fact that boreholes often under sample steep features is worth mentioning in this manuscript.

L474-475: I remain to be fully convinced that these populations can't reliably be described using a log-normal distribution, however, can see the usefulness and power in using the Power-law. I wonder whether because many of the controls that underpin the log-normal distribution (e.g. mechanical stratigraphy) are scale dependent (e.g. formation, facies, bed, sub-bed, lamination), whether these are masked within a powerlaw relationships scatter. It may well be difficult to sample many scales of mechanical stratigraphy, so maybe the authors approach is the best we can do to inform our modelling.

L481-483: Some of the local slope variability also appears to be due to sampling bias due to the orientation of the scanline, extent of exposure, and orientation of the fracture.

L512: I would be fascinated to see how this translates to some of the datasets I've been working with in the Carboniferous. I suspect the strong mechanical stratigraphy will cause several scale-dependent log-normal populations depending what mechanical facies the fracture intersects (and offset for faults).

SED
Section 8.2: I feel you are missing a key point by raising what is the norm for modelling in fracture controlled plays. The lack of geological input data means that even with several limitations, this work is highly valuable in particular in mature basins. I think the 'purpose' of this paper would be strengthened by weaving this into the introduction and discussion, and potentially highlighting how poor fractures are sampled in the subsurface.

L542: very interesting! Do you have a handle on how much the 'effective' aperture is left after the propping material is discounted? This would provide a more reasonable figure to inform fracture modelling in these lithologies.

Discussion and Conclusions: I feel that these sections could be slimmed down a bit and the paper loses a little focus. Overall, however, it is clear the importance of studies like this in forwarding our ability to predict fluid flow in fracture dominated plays.

Fig1: Overall, the figure could do with a little redrafting. the arrows to the censored fractures appear too much like fractures in panel (a), consider changing the colour? In panel (b) the title 'circular scanline window' should in fact be 'circular window' as no scanline analysis is shown (the scanline refers to the counting of 'n-nodes' and represents a 1D technique). The picture for Box counting could be increased to better illustrate the technique.

Fig2: consider adding 'u' nodes to the censored fractures on panel (d) and a branch triangle should be included.

Fig3: A zoomed out scale map should be included for non-UK based readers who may not be familiar with northern Scotland. The attribution on panels a) and c) is unclear and should be typed out as apposed to a screen shot. Panel C) misses it's panel label and a north arrow. In panel C it appears that there are two fracture, one of which is censored at a different trace-length compared to the other due to the exposure and orientation of the coastline relative to the fracture sets. The weathering patterns on (d) appear to show that the linear scanline trends sub-parallel to a set of penetrative fracInteractive comment

tures that run sub-parallel to the post. If this is the case then this set will be drastically under sampled leading to a lower trace-length that the true population. The scale in e) and f) are unclear and f) misses it's panel label.

Fig4: On panel j) IC and CC are switched over.

Fig6: Upon looking at these curves it appears many populations are sampling more than one trend. Using the example of the sub-regional length, the jump in the cumulative frequency curve just prior to 102 appears to match the exposure distance of one of the fracture sets in Figure 3c. I would be very interested to see the data by sets that's underpins this work, as I suspect some of this variability is due to the sampling of multiple populations.

Fig7: I feel this figure would be best in the supplementary, with an example included as part of a figure in the main text that incorporates figures S2 and S3. Should fig7 stay, then the y-axis in panels a) and c) need labelling and the text in panel c) and potentially b) is too small to read clearly.

Fig8c: I would suggest labelling the aperture and trace length trends in the figure for clarity.

Fig9: This figure presents some very interesting data, and raises a few questions. 1) is the regression co-efficient a combination of both veins and 'other' (I assume joints and faults are grouped here), as there appears to be a large clustering of off-trend features in the other category. While I would agree that veins display a strong trend, I am less convinced by the joints and faults. Particularly when looking at the small aperture results (-5) which range roughly from log length -1.4-0.4 m. I agree that there has to be an upper bound, and that this will increase as length increases, however the lower bound is much harder to suggest. Is it a function of aperture can be larger for larger unfilled fractures, but large unfilled fractures don't need to have large apertures. This makes sense as open voids that are not propped can easily be closed during later tectonic events, and faults are known to be able to form with very thin fault cores. Is it

SED
really appropriate to combine both these datasets?

Fig10: a) are the fracture traces that are interpreted as terminating at what looks to be a wedge of cover truly i-nodes, or do they continue underneath? In fig. 12 you mention it is sand and gravel, if this is recent cover then I suggest that the interpretation boundary be extended around this and i-nodes not be interpreted here. The name of the headland just above 'Fig. 11a' is very small and difficult to read. What is the scale for rainbow depth scale? B) A number of potential 'subjective bias' issues are present in this interpretation as outlined in my major comment. Also how penetrative are these fractures? Do they extend to the base of the bedding plane? Brackets are missing after the number of nodes in the key.

Fig11: What is the scale for the node intensity? I-nodes also appear to be present along the edge of the ruler?

Fig12: Very interesting data in the topology plots! Great idea to look at different size circles like this and I would be interested to discuss what you found to be best to provide a) a representative connectivity and b) capture the heterogeneity of the system.

Fig14: Nice schematic, is the bedding sub-horizontal in the area?

Table2: you have split the data between joints and non-filled fractures, how where these differentiated? For your ranges you provide the minimum and maximum, however, this gives no reference as to whether these are outliers. I suggest including either mean (if normally distributed) or median values, and potentially also a measurement of variability. This way it could be deduced whether you had a 'characteristic ' length and aperture within a specific scanline. In the supplementary information it would also be good to see the spread of this data by sets.

References

Andrews, B.J., Roberts, J.J., Shipton, Z.K., Bigi, S., Tartarello, M.C. and Johnson, G., 2019. How do we see fractures? Quantifying subjective bias in fracture data collection.
Solid Earth, 10(2), pp.487-516.

Andrews, B. J., Shipton, Z. K., Lord, R., and McKay, L. (accepted, in revision). The role of pre-existing jointing on damage zone evolution and faulting style of thin competent layers in mechanically stratified sequences: a case study from the Limestone Coal Formation at Spireslack Surface Coal Mine, Solid Earth Discuss., https://doi.org/10.5194/se-2019-202.

Laubach, S.E., Eichhubl, P., Hilgers, C. and Lander, R.H., 2010. Structural diagenesis. Journal of Structural Geology, 32(12), pp.1866-1872.

M Mauldon, M., Dunne, W.M. and Rohrbaugh Jr, M.B., 2001. Circular scanlines and circular windows: new tools for characterizing the geometry of fracture traces. Journal of Structural Geology, 23(2-3), pp.247-258.

Nixon, C.W., Sanderson, D.J. and Bull, J.M., 2012. Analysis of a strike-slip fault network using high resolution multibeam bathymetry, offshore NW Devon UK. Tectonophysics, 541, pp.69-80.

Peacock, D.C., Sanderson, D.J., Bastesen, E., Rotevatn, A. and Storstein, T.H., 2019. Causes of bias and uncertainty in fracture network analysis. J. Nor. Geol.

Scheiber, T., Fredin, O., Viola, G., Jarna, A., Gasser, D. and Łapińska-Viola, R., 2015. Manual extraction of bedrock lineaments from high-resolution LiDAR data: methodological bias and human perception. GFF, 137(4), pp.362-372.Shipton et al., 2019

Shipton, Z.K., Roberts, J.J., Comrie, E.L., Kremer, Y., Lunn, R.J. and Caine, J.S., 2019. Fault fictions: systematic biases in the conceptualization of fault-zone architecture. Geological Society, London, Special Publications, 496.

Wilkins, S.J., Gross, M.R., Wacker, M., Eyal, Y. and Engelder, T., 2001. Faulted joints: kinematics, displacement–length scaling relations and criteria for their identification. Journal of Structural Geology, 23(2-3), pp.315-327.
Wilson, C.G., Bond, C.E. and Shipley, T.F., 2019. How can geologic decision making under uncertainty be improved?. Solid earth.

Please also note the supplement to this comment: https://www.solid-earth-discuss.net/se-2020-15/se-2020-15-SC2-supplement.pdf

SED
Fig. 1. Areas of increases subjective bias on the Circual window

---

## Referee Comment (RC2) · Vincent Heesakkers (Referee) · 26 Apr 2020

Comments on MS: "Fracture attribute scaling and connectivity in the Devonian Orcadian Basin with implications for geologically equivalent sub-surface fractured reservoirs" Dichiarante et al.

Referee: Vincent Heesakkers

The paper in question covers a topic that, as the MS mentions, significantly adds to the scientific community by covering multi-scale observations of fracture aperture and frac-

ture lengths in order to establish a scaling laws for fracture attributes over several order of magnitudes. I agree with the authors that there is a need for additional studies like this in order to assist sub-surface modeling of fracture systems, which often is confined to limited wellbore data, covering a given scale. In order to reduce the uncertainty in sub-surface fracture models, and scale wellbore observations to reservoir scale models, outcrop observations are critical. This MS covers outcrop observations of fracture attributes with regional and micro scale to establish scaling laws that can be used in a similar geological setting. Direct use of the scaling laws in this MS might be limited to reservoirs with key geological similarities, after one establishes that the observations in this MS are a reasonable analogue. However, the thorough study of fracture attributes at multiple scales, and the establishment of scaling laws over 8 orders of magnitude, is an example of techniques that hopefully will be used more often within other geological settings. The topic, workflow and findings in this MS are therefore very suitable this journal, and a significant scientific contribution.

The MS is very well written with minor grammatic errors. The technical organization of the MS is properly ordered, with clearly stated problem statements and scientific methods. The results are presented clearly and straight forward, and derived conclusions match the observations stated in the MS. I support acceptance and publication of this MS, as I deem it suitable for this journals special edition.

I have enjoyed reading the MS and do see tremendous value in its study and utilization of its observations and conclusions. Very well organized and well-written.

I do have several general comments/suggestions about the MS listed below, followed by more specific comments referenced to line items in the MS. Hopefully, my comments / suggestions will further improve the quality of the MS.

General comments:

1) There is a general lack of guidance on specific use of the observations and conclusions of the scaling laws for sub-surface reservoir models: results in fig 8 span multiple

orders of magnitude (8), but with a degree of uncertainty. Fig 13 attempts to address these points, but the text is lacking a discussion on utilization of these findings for sub-surface reservoir scale fracture modeling. The authors did state that this is one of their main drivers for this study, thus a more elaborate discussion is warranted.

2) Although specified within the MS, confusion remains on how faults and large lineaments (regional scale) can be used to infer length / aperture in subsurface or micro scale. More discussion is needed. I would have liked to see the analysis of shear fractures separate from opening mode fractures (joints). It gives the impression that, if we can map faults from seismic, we can infer the attributes of joints system. Often the two are not coupled as faults do form associated fracture systems within their damage zones, but this happens independently from pervasively distributed joint systems throughout the reservoir. More discussion needed on why the shear and opening mode fractures are treated similarly over multiple scale of observations.

3) Each fracture system is unique. Even fracture systems in similar host rocks and tectonic regime could vastly differ in their attribute distribution, based on local variations of geological factors. The fracture systems reflect details in the geological hysteresis, and are sensitive to local variations of many geological aspects: local stress field, pore pressure evolution, chemistry, strain rates, diagenesis, geochemistry, etc. The claim the Caithness outcrop is a valid analogue for sub-surface fractured reservoirs like Clair Field is a fair statement, and I don't disagree. But it needs a bit more attention to understand the differences and similarities. The MS tries to convince the reader that the analogue is appropriate, but minimal evidence is provided on why that claim has been made.

In addition, the MS would benefit from a discussion on outcrop vs sub-surface fracturing processes in general. Outcrops often are saturated with fractures, as existing fracture systems get enhanced (saturated) during exhumation processes. In this example, I expect the effect of enhancement due to exhumation to be significant at the studied coastal sections. It deserves a discussion on what assumptions have to be made to

assume the outcrop dataset and its scaling relationships over multiple scales, are valid to use in sub-surface modeling efforts. To apply this technique for a different reservoir, one would need to find compatible outcrops that allow a similar sampling of micro / meso / regional scale fractures as analogue for the reservoir. That might be a difficult task. The MS would benefit from a more elaborate discussion on the use of outcrop analogues for sub-surface fracture systems, and guidance on how the resulting scaling relationships can be used.

4) Paper is missing a discussion on what is assessed as the "length" of the fractures measured at different scales. The length of a fracture as it grows, is different then the lengths that defines fluid flow pathways through connectivity. There is a scale dependency of observation here, that drives the measured length. For example, a single trace on the bathymetric data (Meso scale) might be mapped as several segments on the Macro scale. For example, quality control of fault maps often utilizes fault length vs displacement profiles to identify faults that are mapped with lengths too long compared to their offset, suggesting the fault likely consists of multiple shorter interconnected segments. A discussion on this during multi-scale analysis of datasets is warranted and currently missing. This could be addressed in the discussion near line 123-128. Suggestion: if available, please show the regional scale fault lengths vs displacement to ensure consistent relationships exist, and thus the proper/meaningful fault lengths are recorded.

5) I am skeptical of the micro scale dataset. It seems like the data was collected from a single sample located within or very close to a major fault (Group 3). This very specific location is likely not representative for the background micro scale fracture set as it was specifically chosen based on its micro-fractured appearance. However, this sample ties the scaling relations in Fig 8 to 8 orders of magnitude. I would have liked to see several samples and thin sections at random locations. The micro-scale data is questionable in terms of expansion of the scaling relations. A discussion around this uncertainty needs to be included.

[Figure]

6) The MS would benefit from an expansion through additional plots / discussions to compare observations to other multiscale studies. (for example, Hooker et al, 2014, and additional references listed in line 62)

Specific comments, referenced to line item:

19) clear separation between FAULT width and FRACTURE aperture. However, in most part of the MS faults and fractures are grouped together in the collected and analyzed dataset. Why discuss them so separately in this abstract?

24) both "fracture and fault" length. . ..be more specific consistently when faults and fractures (opening mode?) are addressed as a group and when not. (at this point the reader has not yet seen the discussion on faults / fractures being addressed as a grouping. Although the MS specifies what is meant with faults and fractures in terms of terminology, there is an inconsistent use of these terms throughout the MS.

37) Please be more elaborate on the "heterogeneous distribution of natural fracture systems". It will be good to inform the reader (through proper references; e.g. Narr et al., 2006) what the cause/implication is of this heterogeneity. Unless familiar with natural fracture systems, the reader might not find it straightforward that fracture systems have an inherent degree of heterogeneity.

62) . . .results from multi-scale sampling. . . I do like the statements made her in the MS to highlight the importance of multi-scale sampling and the fact that most outcrop studies focus on given scales. However, a discussion might be warranted here on what previous multi-scale studies have found and how this is different from this MS attempts.

78) . . ., 2019), we do (add comma)

80) fracture attribute (remove "s")

82) In most reservoir models, only a portion of the fracture system in modeled that is responsible for fluid flow to address the primary production of the reservoir. I would add the importance of this study to "secondary recovery mechanisms" and studies

associated to that, as the entire fracture system is of importance when injecting to improve sweep efficiency.

101) correct sentence but the 5 commas on one sentence is confusing. Please restructure the sentence.

105) Reference the Terzaghi method for completeness.

131) Did you mean "Ideally, the best-fit in a power-law distribution at given scale, should be consistent over several orders of magnitude."

177) Suggestion: move geological setting to Ch 2, so that the outcrops are described before the method of data collection. During the review I have gone back and forth on this, as there are pros and cons to moving it to Ch 3 as currently in the MS.

189) More discussion needed on similarity and difference of outcrop as analogue. The different tectonic setting can have a major impact. Again, I don't doubt the statement that this is a good analogue. But more detail here on details why it is a good analogue or why it might not, is necessary.

258) did the field verification add any information on the actual lengths measured on images vs lengths measured in field? This comment goes back to the influence of the scale of observation to fault/fracture length.

279) What is the evidence/observations for the statement that the sets were both active at the same geological event?

290) Single sample, chosen based on its micro-fractured appearance, within (or near) one of the major faults. I am skeptical if this Sample is a good representation of the micro-scale to extend the scaling relations, or if this sample is only representative for sections within or near faults. More samples, chosen at random locations would have strengthened the MS.

389) is this an effect of exhumation processes that have enhanced fracture apertures

of open fractures?

467) How does the exponent compare to other datasets that underwent box counting? What is the meaning of a similar exponent? Is there a physical process behind this or is this a specific character for a fracture system, confirming we are examining fractures of the same population?

568) 14cm aperture, and likely larger exist but difficult to sample with core as the change to intersect is very very slim.

602-612) Confusing terminology: refer either to fracture or fault, but be consistent

634) replace "faults" with fracture

664-665) Figure references: 11a and 11b

672) remove this section from the conclusion. Its location suggests this is a major part of the MS, but its discussion in the text does not warrant that.

Figure 3: Missing labels c and f

Figure 8: To me this is the main figure of the paper. I suggest enlarging figures a) and b) to a larger size as its challenging to read all the "extra" info in the plot. They deserve to be bigger.

---

## Author Comment (AC2) · 15 Jul 2020

We thank Dr Andrews for his supportive comment. We are pleased that he found our study of clear importance and of relevance to readers of Solid Earth. We address in this reply his major comments and we have taken his minor comments on board in our extensive revisions to the manuscript. We have reorganised the manuscript and introduced new material to support our assumptions and discussed the limitations much more explicitly.

[Figure]

1) The analysis of several sets and fracture types within the same population.

This point has also been raised by both reviewers. We have explicitly addressed this issue in our revised paper in the following ways 1) We make it clear that the attribute analysis is focused on the Group 3 structures identified by Dichiarante et al (2016). These include both faults and opening mode strcutures where we observe them in outcrop. At our Dounreay location, faults with metre-scale contained the same mineralisation as opening mode fractures. They clearly contributed to the flow in the subsurface. In fact they are also conduits for meteoric/groundwater fluid flow in many cases as these faults are wet and the surrounding rocks are dry. 2) In the discussion we make it clear that the assumption that the extent to which the scaling of fracture aperture attribute from macro-scale to the regional scale structures needs to be tested.

2) Subjective bias during the digitisation of fracture traces We thank the reviewer for highlighting this aspect of the study and drawing our attention to his interesting paper. We accept the point that variations in the exposure and the presence of the ruler in the photo will have created a bias in the results. We have added some discussion of this aspect to section 6 in our paper.

3) Power-law versus Log normal The reviewer raises a very good point. We agree that the power law slope and intercept values could reflect variations in lithology, proximity to major structures and other aspects that would be relevant to producing a geo-model. We don't have enough data in this study to say much about this but would draw attention to a recently published Open Access paper (McCaffrey et al 2020, J. Geological Society of London) in which we reported over 100 fracture datasets in basement lithologies. In this much larger study, we show that proximity to major structures produces an increase by more than an order of magnitude in fracture intensity (y-axis intercept) for aperture data. Differences in the scaling (power law exponent) we attribute to different preservation levels below regional unconformities. There is clearly more work that can be done on this aspect.

---

## Author Comment (AC3) · 16 Jul 2020

In response to reviewer 2's numbered points and general comments.

| | |
|---|---|
| 1) There is a general lack of guidance on specific use of the observations and conclusions of the scaling laws for sub-surface reservoir models: results in fig 8 span multiple orders of magnitude (8), but with a degree of uncertainty. Fig 13 attempts to address

these points, but the text is lacking a discussion on utilization of these findings for subsurface reservoir scale fracture modeling. The authors did state that this is one of their main drivers for this study, thus a more elaborate discussion is warranted. | 1)It was not our aim to provide guidance on the specific use of the observations. We aim to show here one way in which the observations might be applied to an example reservoir (we used Clair). We have revised the introduction and discussion and hope this is now clearer. |
| 2) Although specified within the MS, confusion remains on how faults and large lineaments (regional scale) can be used to infer length / aperture in subsurface or micro scale. More discussion is needed. I would have liked to see the analysis of shear fractures separate from opening mode fractures (joints). It gives the impression that, if we can map faults from seismic, we can infer the attributes of joints system. Often the two are not coupled as faults do form associated fracture systems within their damage zones, but this happens independently from pervasively distributed joint systems throughout the reservoir. More discussion needed on why the shear and opening mode fractures are treated similarly over multiple scale of observations. | 2)We have explicitly addressed this issue in the revised paper in the following ways

A)We make it clear that the attribute analysis is focused on the Group 3 structures. These include both faults and opening mode strcutures where we observe them in outcrop. At our Dounreay location, faults with metre-scale contained the same mineralisation as opening mode fractures. They clearly contributed to the flow in the subsurface.

B) In the discussion we make it clear that the assumption that the extent to which the scaling of fracture aperture attribute to the |

| | regional scale structures needs to be tested. |
|---|---|
| 3) Each fracture system is unique. Even fracture systems in similar host rocks andtectonic regime could vastly differ in their attribute distribution, based on local variations of geological factors. The fracture systems reflect details in the geological hysteresis, and are sensitive to local variations of many geological aspects: local stress field, pore pressure evolution, chemistry, strain rates, diagenesis, geochemistry, etc. The claim the Caithness outcrop is a valid analogue for sub-surface fractured reservoirs like Clair Field is a fair statement, and I don't disagree. But it needs a bit more attention to understand the differences and similarities. The MS tries to convince the reader that the analogue is appropriate, but minimal evidence is provided on why that claim has been made. | 3) We have addressed this directly in a new section (2.2) that has been added to the paper during the reorganisation which explains clearly the basis for the choice of analogue. |
| In addition, the MS would benefit from a discussion on outcrop vs sub-surface fracturing processes in general. Outcrops often are saturated with fractures, as existing fracture systems get enhanced (saturated) during exhumation processes. In this example, I expect the effect of enhancement due to exhumation to be significant at the studied coastal sections. It deserves a discussion on what assumptions have to be made to assume the outcrop dataset and its scaling relationships over multiple scales, are valid to use in sub-surface modelling efforts. | We only measure the Group 3 faults and fractures that we know formed in the subsurface before exhumation |
| To apply this technique for a different reservoir, one would need to find compatible outcrops that allow a similar sampling of micro /meso / regional scale fractures as analogue for the reservoir. That might be a difficult task. The MS would benefit from a more elaborate discussion on the use of outcrop analogues for sub-surface fracture systems, and guidance on how the resulting scaling relationships can be used. | Finding compatible outcrops is a good point in applying this method elsewhere. It is out of scope for this manuscript to review the use of outcrop analogues for sub-surface fracture systems. That is a very large topic. |
| 4) Paper is missing a discussion on what is assessed as the "length" of the fractures measured at different scales. The length of a fracture as it grows, is different than thelengths that defines fluid flow pathways through connectivity. There is a scale dependency of observation here, that drives the measured length. For example, a single trace on the bathymetric data (Meso scale) might be mapped as several segments on the Macro scale. For example, quality control of fault maps often utilizes fault length vs displacement profiles to identify faults that are mapped with lengths too long compared to their offset, suggesting the fault likely consists of multiple shorter interconnected segments. A discussion on this during multi-scale analysis of datasets is warranted and currently missing. This could be addressed in the discussion near line 123-128. Suggestion: if available, please show the regional scale fault lengths vs displacement to ensure consistent relationships exist, and thus the proper/meaningful fault lengths are recorded. | We do not have displacement data for the regional datasets as these are mapped on imagery. We have added some discussion of this limitation. |

| | |
|---|---|
| 5) I am skeptical of the micro scale dataset. It seems like the data was collected from a single sample located within or very close to a major fault (Group 3). This very specific location is likely not representative for the background micro scale fracture set as it was specifically chosen based on its micro-fractured appearance. However, this sample ties the scaling relations in Fig 8 to 8 orders of magnitude. I would have liked to see several samples and thin sections at random locations. The micro-scale data is questionable in terms of expansion of the scaling relations. A discussion around this uncertainty needs to be included. | We acknowledge that this is a limited dataset but we include it as it provides an upper limit on intensity values at this scale. We have added discussion of this in the paper. |
| The MS would benefit from an expansion through additional plots / discussions tocompare observations to other multiscale studies. (for example, Hooker et al, 2014, and additional references listed in line 62) | We have added discussion of Hooker et al (2014) |

Reviewer 2 - Line by Line comments

| |
|---|
| 19) We have revised this sentence and removed the need for the distinction |
| 24) We have revised the abstract to make the terminology consistent with how we use it throughout the paper. |
| 37) We have completely rewritten the introduction with many more references and a more nuanced discussion of the causes of fracture heterogeneity. |
| 62) We have added more discussion of previous multi-scale studies |
| 78) Section has been rewritten |
| 80) Section has been rewritten |
| 82) We have added this point to the revised introduction. |
| 101) We have simplified this sentence |
| 105) We have added this reference and cited it at this position in the text. |
| 131) No – but we have clarified this sentence in our revisions. |
| 177) We accept the reviewers point and have reordered the sections – placing Geological Setting before the methodology. We have reordered the figures appropriately. |
| 189) A new Section 2.2 has been added to cover this point. |
| 258) No. the fault lengths cannot be verified in the field due to the limitations of the exposures. We have clarified that we mean here the lineaments were verified to be natural features and not anthropogenic |
| 279) See Reviewer 1 reply. We have clarified the evidence for this. |
| 290) See reply to Reviewer 1 – we have added 3 sentences to acknowledge this limitation |
| 389) Because we are measuring kinematic aperture these fractures are not 'open'. We are quite convinced the apertures have not been affected much by the exhumation process as they formed in the subsurface and were partially to wholly mineralised. We purposely avoided joint systems as these may well be exhumation related |

| |
|---|
| 467) We have added some text to the revised version to explain the significance of the box counting dimension and say that the values obtained are in line with previous studies. The box counting dimension is generally not sensitive enough to distinguish between different fracture populations unless the are radically different. |
| 568) As stated in the text this was observed and recorded by Franklin (2013) |
| 602-612) We have replaced 'fault' with 'structure' in this paragraph |
| 634) Done |
| 664-665) corrected |
| 672) We accept this point and have deleted this section |
| Figure 3) corrected missing labels |
| Figure 8) We have made this figure bigger |

---

## Author Comment (AC4) · 16 Jul 2020

Reviewer 1 General comments and suggestions in red

| Comment and suggestion | Reply/Action |
|---|---|
| The main claim of the paper is that it is possible to combine patterns for all types of fractures (opening-mode fractures and faults) imaged from micro-scale to regional scale to find meaningful size scaling patterns. Another claim is that such broad scale scaling observations can be used by making projections to other scales of interest to get input for or to provide information for 'realistic' reservoir models and as input for fluid flow simulations, etc.
There is room for improvement in how clearly these claims are stated, how they are related to previous work, and how well they are defended and supported. | We have revised the manuscript extensively taking into account the reviewers comments with the aim of clarifying the claim and how our data support that claim. We are also now much clearer about the limitations of our study. |
| 1(a)The Introduction should have a clearer statement of claims.
The paragraph should be broken up to separate the inventory part from a revised and augmented section that explicitly spells out the claims; text that could start out 'Here we show that. . .'. A clear statement of the claims is essential.
1(b) These claims also need to match the Conclusions. Neither of these conditions are currently met. The text doesn't make the claims clear.
1(c) And the first conclusion (line 640) that the outcropping rocks are a 'direct analogue' is not a conclusion at all. This point was merely asserted in the text without much back up. The comments in line 604-605 seem to point just to a similarity in spacing values. But if this is a major conclusion it needs to be signaled more clearly and the evidence needs to be presented more effectively.
It may be easier to just assume that the outcrops may be pretty good analogs and present the evidence for this without making it a major conclusion (but explain what you are doing). | a) We have completely rewritten the introduction to make the claim clear as the reviewer requested.

b) The conclusions have been rewritten to match the claim

c) we have removed the first conclusion and followed the reviewer's recommendation. |

| | |
|---|---|
| 2(a) The discussion of previous work is not adequate. The account needs to be more complete and more nuanced.
Some of the relevant references are cited, but the scope of this previous work is not clear from the presentation. A more informed and complete account is needed.
2(b) Also missing are some of findings from previous work that bear on the main claims of this MS. For example, large aperture size distribution data sets for opening-mode fractures have been collected from a wide range of sandstones.
(for example, Hooker et al. 2014) with the aim of predicting the average spacing/intensity of open fractures in reservoirs, and some of these predictions have been tested with horizontal cores or outcrop analogs. This previous work needs to be accounted for more explicitly. And providing a comparison of the results in the current study to the findings of Hooker et al. (2014) seems like an obvious step for putting the current work into context. It should also be addressed in the paper that extensive size-scaling investigations show that some sandstones do not show wide fracture size scaling ranges (see next item).

The limitations of scaling that have been found need to be acknowledged. Some tests like Hooker et al., 2009 show that, of example, microfracture aperture size distributions can be projected over several orders of magnitude to accurately predict intensity at sizes where fractures can impact production. But other studies, for example Laubach et al. 2016, show that in some sandstones, fractures have a narrow (characteristic) aperture size range and accurate projections from populations of small aperture sizes to large are impossible. This raises a concern that is directly related to the claims of the MS, since these observations imply that some fracture patterns do not scale (they don't have scale invariant properties; they can't be projected to or from larger or smaller sizes). What about circumstances where there is evidence of narrow fracture attribute size ranges? The evidence of the literature seems to be saying that some fractures patterns 'scale' but others do not. Taking a for instance from within the area represented by MS figure 3, Laubach et al. 2014 J. Struct. Geol. showed that two adjacent sandstones, influenced by faults or the same population as described in this MS, have drastically differing fracture attributes (size, spacing, porosity preservation). According to the proposition in this MS, these attributes should be predictable by the MS's regional scaling relation. The test case should be discussed. It's hard to see how the regional could get this right, since these contrasting patterns are on the same. scale. But the differences between the two sandstones are just those that would affect reservoir behavior. The MS claims on this topic need to be reconsidered or at least more completely explored in this light. | a) We have added much of the literature that the reviewer has provided. The introduction is now more complete and nuanced.

b) we have included the Hooker et al (2014) findings in our introduction and in our discussion.

We have added discussion of this. We are not saying we have found a general law – that all fractures within sandstones show a wide range of scaling but we are saying that in this instance – the Group 3 structures in the Devonian rocks do show scale invariant properties and this is a good analogue for Clair. |
| 4. A related problem is in the description of studies that examine fractures having a wide range of sizes. The contrast between 'given scales' versus 'multiscale' is problematic, since 'given scale' seems to imply a narrow size range, but some of the studies cited under 'given scale' cover three or four orders of magnitude in scale. Maybe this is just an oversight. The types of structures analyzed and the size ranges analyzed need to be accurately portrayed. Moreover, since the outcrop structures in the reservoir sandstone analog in this MS seem mainly to be opening-mode fractures, the MS should pay closer attention to the previous work on scaling of opening-mode fractures in sandstone. It's surprising that there is no explicit comparison with the compendium of data in Hooker et al. (2014) for example. Or any discussion of the problems with collecting reproducible length data in sandstones outlined by Ortega and Marrett (which is in the reference list). | We have addressed this issue in the new introduction to the paper |

| | |
|---|---|
| 5. Is the distinction in this MS between 'given scales' versus 'multiscale' between data sets where the structures are clearly genetically related and of the same type, versus mixed populations of opening-mode fractures and faults that may not be related? The text I think could be read this way although this isn't stated explicitly. This part of the MS may be the most problematic. As noted in the comments below keyed to lines in the text, it is not always clear what kind of structure is being compared or projected. This needs to be corrected. Partly this problem in the text comes from using the general term 'fracture' to mean either opening-mode fracture or fault. This usage is stated right at the outset. But it leads to problems, confusing and obscuring the argument. The case is being made in the MS seemingly that, for example, patterns of faults visible on seismic can be used to predict the size distributions and connectivity of opening-mode fractures at the reservoir/outcrop scale. This is a very considerable claim (I'm dubious). But the claim should at least be made explicitly and defended openly. | We have explicitly addressed this issue in the revised paper in the following ways 1)We make it clear that the attribute analysis is focused on the Group 3 structures. These include both faults and opening mode strcutures where we observe them in outcrop. At our Dounreay location, faults with metre-scale contained the same mineralisation as opening mode fractures. They clearly contributed to the flow in the subsurface. 2) In the discussion we make it clear that the assumption that the extent to which the scaling of fracture aperture attribute to the regional scale structures needs to be tested. |
| 6. The claim that multiscale analysis can be useful for informing geological models has been supported by examples from the literature (these should be noted) but **the claim that, for example, regional lineaments and seismically detected fault trace patterns can be used to predict meaningful fracture attributes at the grid block** or smaller scale seems to me to be a bridge too far. If this is the claim, then a more convincing case is needed to support it. | We have discussed more example of multiscale analysis and we discuss the application of this more carefully in the the discussion section |
| 6b An obvious concern is the projections in figure 8. This figure seems to be saying that aperture and length can be predicted to within two orders of magnitude. What are the error bars on that already really wide prediction? How could such a prediction be used? The authors need to explain how to be useful, 'predictions' can span orders of magnitude (compare the prediction of Hooker et al. 2009 with the two orders of magnitude of size range in the projections of figure 8). Core and outcrop analog data show that fracture patterns at the core and outcrop scale can vary considerably in ways that directly impact fluid flow. As noted above, with adequate samples where microfracture populations are present some of these attributes can be accurately projected over three or four orders of magnitude to predict the attributes of large fractures. But these are cases where the small and | We have not calculated error bars on this prediction – we wish to limit ourselves here to showing the concept of how length scaling constrained over 8 orders of magnitude could be used to estimate apertures. |

| | |
|---|---|
| large fractures are growing and interacting together in a specific rock type. Contemporaneous, interacting fractures are the ones likely to develop power-law size distributions (Cladouhos, Marrett 1996). | In Figure 8 we plot schematically illustrate all the datasets to show the range in intensities from different exposures to illustrate the inherent variabilities. We have modified the discussion of this plot in the revised version and schematically added some low and high strain power law lines to Figure 8c. The C & M 1996 paper is interesting and we have included it. |
| 7. The referencing of certain points needs to be made more complete or more accurate. I've flagged instances in the following detailed notes. As it stands now, I don't think the MS properly represents or credits previous work. | We have added many of the references the reviewer has suggested. |
| 8. There are a number of places in the text where reorganization is needed. The Introduction could be clearer. Some of the material in the Discussion looks more like observations/results. I've flagged some of these issues in the detailed line comments that follow. Improving the overall presentation will increase the impact of the paper. | We have reorganised and rewritten the manuscript extensively in line with the comments form the reviewer. |
| 9. Meaning I've flagged some areas in the text where meaning is unclear. | We have tried to improve the meaning where it has been flagged. |
| My opinion. This broad statement about fault and opening-mode fracture size scaling is true to an extent. Marrett et al. (1999) documented power-law scaling across 3 to almost 5 orders of magnitude regardless of rock type or movement mode. This was the study that established that such systematic relations exist and that extrapolation from one scale to some other scale of interest was a feasible approach. It's a surprising omission to leave this paper out. One thing that Marrett et al. did not do was to mix opening-mode fracture and fault data sets. Doing so requires some defending. It's ok to make the general point that some faults and some opening-mode fracture populations show scaling patterns (although subsequent work shows that some populations do not scale in this way). But it is problematic to lump them all together as 'fractures' if in your description and discussion you let the reader lose track of which kind of structure you are talking about. You are making the claim that it doesn't matter which type of 'fracture' is analysed - that's fine if you can defend it but it's not convincing if you just use the all-purpose word 'fracture' in a way that makes it hard for the reader to assess the | We thank the reviewer for reminding us about the Marrett et al 1999 study which we now include.

See above reply to point No. 5 |

strength of your claim. For example, in lines (602-612) it's hard to tell which type of structure you mean. Marrett, R., Ortega, O. J., & Kelsey, C. M. (1999). Extent of power-law scaling for natural fractures in rock. Geology, 27(9), 799-802.

Reviewer 1 – Line by line comments (numbers refer to original manuscript)

| |
|---|
| 34-37 We have rewritten the introduction taking into account this feedback and providing a greatly expanded number of references |
| 37. We have added the de Dreuzy reference and included this general point |
| 38. We have revised this section |
| We have clarified this in the rewrite |
| 42. We have clarified what we mean here in the rewrite |
| 43. This is a good point that we omitted and now include specifically and cite the Laubach et al 2019 paper. |
| 44. Our study was completed before the JSG Special issue was published but we have now added the result a spatial correlation analysis to the manuscript as plots in the supplementary file and discuss the results. |
| 48. We have rewritten this section and removed any impression that opening mode fractures can be detected on seismic. Interestingly, elsewhere we have published work on opening mode fissure structures in basement rocks that potentially are expressed in seismic attribute maps of basement highs (the Lancaster field). We do not mention it here as the lithology is different. |
| 50. We agree with this point and in our re-writing of the introduction have taken these comments into account. |
| 52. This has been corrected |
| 54-63. This has all been rewritten |
| 56. We have included the Ukar et al study in a new section 2.2 which discusses the validity of the Orcadian basin analogue for Clair |
| 57. We have revised the introduction so this point np longer applies. |
| 62. We have included these studies in our introduction |
| 67. We have added a new section 2.2 which discusses this aspect directly |
| This is a valid point that we acknowledge in our discussion |
| 75. We have clarified this |
| 76. We have clarified that we do mean aperture and not fault width for these structures. |
| 77. We have rephrased this sentence |
| 80. We have edited this sentence |
| 89. We have added the reference to Marrett et al 2018 and changed 'spacing' to 'spatial arrangement' |
| 103. No. we used a feeler gauge which is used in engineering to measure widths as small as 0.02 mm. We used this tool in conjunction with a hand-lens in the field or on a hand-sample. We have modified the text by adding 'an engineering feeler gauge in conjunction with a hand lens' to make this clearer. |
| 113. We accept this point and have made it clear in this paragraph that spacing attribute is more likely to show log-normal distribution whereas size attributes are more likely to show power-law scaling. |
| 125. We have removed 'although unknown' from this sentence to avoid making this inference |
| 127-132. We have edited this section carefully to make the point suggested here about limits to power-law behaviour. We have also made it clear in the following section that power-law scaling should not be assumed |
| 132. This was an error – we have corrected this to 'over several orders of magnitude length scale'. We moved this sentence as part of revisions in response to the previous comments |
| 151. We recorded fractures that are related to the Group 3 set – this is made clear in the 'Geological Setting' section |
| 156. We have changed this to 'box-counting dimension' – which is the plot that we are referring to here. |
| 157. Correct – we have added 'of connection types' for clarity |
| 165-175. We agree and have added 'fracture trace' before connectivity to make this clear. |

| |
|---|
| 188. We have added a a new section (2.2) justifying the use of the Orcadian basin as an analogue for the Clair reservoirs and included the Ukar et al criteria. |
| 210. The point here is that these fracture fills show the structures formed in the subsurface and are not related to exhumation of the analogue. We have clarified this point in the text. |
| 233. Yes and we have moved text that described the evidence for this in the discussion to this part as we realize now it is important primary evidence |
| 256-261. We added an estimate of the difference this makes to the number of fractures we were able to record as follows. |
| 279. We modified the text to say ….'they mutually cross-cut each other which enabled us to infer that they were active during the same geological event' to make it clear what our evidence was. |
| 291. We added text to explain this and acknowledge the limitation. 'At the scale of a thin section, only samples from fault zones contain enough fractures to produce a statistically significant sample. We thus recognise that the results at this scale are representative of fracture intensities within fault zones and provide an upper limit relative to background. At the scale of a thin section, only samples from fault zones contain enough fractures to produce a statistically significant sample' |
| 285-299. See above comment |
| 314. We have analysed the spatial arrangement using the Marrett et al (2018) method and have added a short discussion of the results and the plots to the supplementary file. |
| 350. We have added a citation to this paper |
| 361. Corrected by removing 'coefficient' |
| 368. Edited this paragraph for clarity. |
| 370 Corrected – removed this sentence in rewrite |
| 380-389. This is now discussed in the last paragraph in section 8.1 |
| 394. We have added the evidence that the features in the bathymetry are the same as those in the adjacent coastline. We also added some context for the Brims Ness photo location as well. |
| 412. We have added a citation to the Questiaux et al references and now report the clustering recorded by the spatial arrangement analysis. |
| 459-463. We have added a sentence in this section which explains the significance of the box counting dimension and referenced Hirata (1989) |
| 465-467. We agree the box counting doesn't tell you anything about the spatial arrangement and as noted above we have performed the spatial correlation analysis and reported the results. |
| 467. As Walsh & Watterson (1993) discussed 'A fracture pattern incorporates many different attributes such as orientation distribution, size population and fracture trace geometry'. There is no simple relationship between the box counting dimension and the size distributions |
| 471-500. We have taken the reviewer's advice an broken this section into 3 paragraphs and state the supporting evidence for the assertions more clearly. |
| 475. We discuss truncation and censoring in Section 3.1.1 and now make it clear again here what the effect can be and that our MLE methods can help to reduce the uncertainty in fitting power-laws to datasets that have a somewhat limited scale range. |
| 481. We have clarified that the intensity variations and slope variability could be due to samples taken in slightly different contexts (e.g. inside or outside a damage zone). |
| 481. We have not been able to access the Hooker et al (2014) data as these are not available publicly. We have added in the discussion that our results are in agreement with Hooker et al (2014) |
| 483. We have clarified what we mean by 'reducing the influence of individual datasets' by rewriting this section. We have justified plotting the data from different scales because we are confident they ARE genetically related as we have explained in section 3. |
| 489. We have corrected the reference to Gillespie et al 1999. We have reported in section 5.1, the results from the Spatial correlation analysis recommended by the reviewer. (we note that most of this work was done prior to the publication of this volume). |

| |
|---|
| 496. We have clarified what we mean here and changes made in preceding paragraph discussed our result in context of Hooker et al (2014). |
| 502. We have given citations to a number of previous studies in a sentence added to this section. |
| 512. We have now cited Olson (2003) in this section and clarified how our aperture data fit with his model. |
| 515. We have added discussion of our 0.65 exponent in terms Olson (2003) and Schultz et al. (2008) as well as a recent study by Mayrhofer et al. Schultz et al. and added citations |
| 520. We have moved this section to section 2 – Geological setting to provide evidence to support the Group 3 fractures being similar to Clair structures |
| 522. We have changed this to make it clear that it is the fractures that contain the fault rocks. We have deleted the 'hydrothermal' form this section as its somewhat irrelevant to point that we are making here – that the fractures and their fills show similar features to Clair. |
| 523. We refer to Dichiarante et al 2016 where these structures are described |
| 534. We refer to Dichairante et al (2016) where evidence for this was discussed. |
| 548. We agree – but have not changed our text as this is covered by 'dramatically reduce' flow |
| 50? We have explained more clearly at the start of this section what we are attempting to do. |
| 551. corrected |
| 553. We have clarified the Coney et al study is based on wells and aeromagnetic surveys. Coney described these as faults sets so we have changed the terminology to this.Yes they are spaced at these intervals |
| 560-561. Yes – if we can use spacing/intensity to predict length then we can use it also to predict aperture given the aperture/length scaling relationship we presented and discussed in an earlier section. We have added discussion of this assumption in |
| 562. We have clarified what we mean here |
| 562. We now discuss this assumption explicitly. In this application, we use the length attribute at regional scale and consider what the 'aperture' would be given the scaling relationship. It is using the assumption that we know from onshore evidence that faults are contributing to the overall subs-surface fluid flow. |
| 567. corrected |
| 568. We have clarified that the structure was found in the core. |
| 563-8. We have now made a new paragraph that explicitly discusses the assumptions and limitations of our approach |
| 574. We have limited the use of fracture corridors to the regional scale – which is what has been suggested by previous authors. We now refer to the mesoscale examples as clusters of interconnected fractures as the reviewer suggested. We have removed the sentence about 'minimum' estimate as this suggestion is not well constrained. |
| 584. We have added into this discussion the insights from the spatial correlation analysis. |
| 582. We have limited the use of fracture corridors to the regional scale where we do mean 'abnormally closely spaced subparallel fractures'. We have define our usage of fracture corridors. |
| 584-587. We have revised this section to make it clear what we mean and restricted our comments to fracture connectivity in 2D. |
| 589-595. We have now report the spatial correlation results and 2D spacing estimate in the results section - see sections 5.1 and 7.1.1 |
| 602. We have reorganised the discussion so that the flow is more logical. |
| 606. Yes |
| 602-612. We have rewritten this section to improve clarity. |
| 608. No these data re not available |
| 607-608. Rewritten to make it clearer |
| 614-625. We hope the rewritten section is now clearer. |
| 616. We have revised this paragraph and included the Philip et al reference – we are grateful to the reviewer for bring it to our attention. |

| |
|---|
| 614-615. We have modified the sentence slightly to change the emphasis away from being economically 'useful' to providing a useful insight into subsurface fracture properties. |
| 619. We have clarified our assumptions and limitations with respect to this in the discussion |
| 628. Added the Primaleon paper which is now published |
| 634-638. Correct. We have removed this text. |
| 640-641. We have toned down the assertion and now describe the results from Orcadian basin then mention the application to Clair |
| 652. Corrected to 'fracture' |
| 673. Drilling strategy is mentioned in the discussion – its more clearly flagged now. Despite the uncertainty the application is still useful. |
| 872. No as Walsh and Watterson pointed out the artefacts arise when large areas of no exposure are included in a fracture pattern. This is not the case in this example |
| 876-878. We have deleted b and c from Fig 1. |
| 876. These are example distributions. We have clarified this in the caption |
| 898. We added a reference to the locations in Figure 3. All observations are from 'target' sandstones. |
| 902. We have removed 'present-day' and replaced with an imposed stress. |
| 909. Replaced with 'kinematic aperture' |
| Figures. As noted above we are not able to compare our results with Hooker et al 2014 as their data are not available. We have made qualitative comparison in the discussion |
| Fig. 2. We have given the alternative classification in the in the figure caption but retain this classification as we find it useful. |
| Fig. 3 We have fixed the labelling and improved the description of the figure |
| Fog. 6. See comment above with regard to Hooker et al. |
| Fig. 8. We have done this in the text. |
| Fig. 9. We have corrected this to 2 decimal places |

---

## Author Response (AR3)

**MS se-2020-15 - Response to Editor**

We have incorporated both editor comments in a modified version of the paragraph - specifically lines 209-218 in the final manuscript. These changes have removed any question of the validity of separating opening-mode and shear fracture. We will save this argument for another day!

Most Group 3 fracture sets are made up of fracture meshes (sensu Hill 1977, Sibson 1996) formed by closely interlinked sets of contemporaneous shear fractures and tensile veins (Dichiarante et al. 2016, 2020). Thus, in each sample, all fractures considered to belong to an individual fracture set (in this case Group 3) were included in the analysis regardless of opening mode. Thus in our view it is not possible in this study to separate brittle structures into separate sets of simple tensile and shear fractures. This practical approach ensures comparability with subsurface structures in Clair cover sequences and related fractured basement studies where similar interlinked mesh systems are dominant (see McCaffrey et al 2020). One reason for the development of such mesh networks is that many of Group 3 structures reactivate earlier (Group 1 and 2) brittle structures and therefore display a variety of hybrid opening modes (Dempsey et al. 2014).